# N-acetyl-glucosamine primes *Pseudomonas aeruginosa* for virulence through a type IV pili/cAMP-mediated morphology transition

Jing Chen [1] ✉, Guiying Lin[1,2,3], Kaiyu Ma[1], Yunxue Guo [4], Zi Li[1], Xiaoxue Wang [4,5] & Dominique Ferrandon [1,2,3] ✉

A microbe is pathogenic when it manages to survive in its host and, often, is able to proliferate. Thus, virulence entails coping with host defenses in parallel to dissemination and/or attack of the host. How the pathogen endures the attack by effectors of the immune response remains insufficiently understood. Here, we report that planktonic *Pseudomonas aeruginosa* is not immediately virulent in a Drosophila model of acute infection. Bacteria undergo a maturation step called priming, which is required for transition to virulence. Primed bacteria switch to a bacillus shape, only in vivo, proliferate and resist the action of a specific combination of antimicrobial peptides. This priming mechanism requires an interplay between two major effectors of the type IV pili (T4P), FimV and Vfr, which enhance lateral cell wall peptidoglycan synthesis. Interestingly, N-acetyl-muramic acid (NAM) abolishes the virulence of the injected bacteria, which become round, and prevents the localization of the T4P hub protein FimV at bacterial poles, and cAMP signaling. In contrast, N-acetyl glucosamine (NAG) counteracts the action of NAM by promoting FimV polar placement. In fact, NAG alone accelerates the speed of *P. aeruginosa* priming in a PilJ-dependent manner. This suggests that the NAG sensed by microorganisms is a common signal that promotes virulence through a morphological switch, both in bacteria and pathogenic dimorphic yeasts.

*Pseudomonas aeruginosa* is a ubiquitous opportunistic pathogen able to infect a variety of hosts from plants to animals[1] and is a significant threat in terms of public health[2,3]. Besides the mechanisms of antibiotic resistance, it is essential to understand all of the facets that make this bacterium a high-risk pathogen. An essential aspect of *P. aeruginosa* biology is its adaptability to a variety of natural and host environments, allowed by some 60 two component sensing systems, four chemosensory systems, including the Chp system of type IV pili (T4P), various quorum-sensing systems and multiple secretion systems that allow the release of various secreted virulence factors, from siderophores to toxins[1,4,5]. A prevailing view of *P. aeruginosa* pathogenicity is that it functions according to two distinct, antagonistic modes. Planktonic cells are cytotoxic, induce strong inflammatory reactions, and cause acute infections. In contrast, following a lifestyle switch, biofilm-forming bacteria are at the origin of chronic refractory infections such as those found in cystic fibrosis patients[6,7]. This view of two pathogenic lifestyles may need to be amended as studies investigate *P. aeruginosa* pathogenicity at the whole organism level in genetic model organisms. An illustration of its flexibility in terms of virulence programs is provided by three types of infection models in *Drosophila melanogaster*[8,9].

[1]Sino-French Hoffmann Institute, School of Basic Medical Sciences, Guangzhou Medical University, Guangzhou, China. [2]Université de Strasbourg, Strasbourg, France. [3]Modèles Insectes de l'Immunité Innée, UPR 9022 du CNRS, Strasbourg, France. [4]South China Sea Institute of Oceanology, Chinese Academy of Sciences, Guangzhou, China. [5]Key Laboratory of Tropical Oceanography, South China Sea Institute of Oceanology, Chinese Academy of Sciences, Guangzhou, China. ✉e-mail: chenjing_1127@qq.com; D.Ferrandon@unistra.fr

In an injection paradigm in which access to the body cavity is provided by bypassing natural physical barriers, *P. aeruginosa* kills its host within about two days through uncontrolled bacteremia[10–12]. In contrast, continuously ingested *P. aeruginosa* lead to the demise of the flies in about a week, again mostly caused by septicemia once the bacteria have escaped from the digestive tract[12,13]. Finally, in a latent infection model in which flies feed on *P. aeruginosa* for a limited period of time, the bacterium is hardly pathogenic even though it manages to silently colonize the internal tissues of the host in the absence of biofilm formation[9].

One potential clue to account for this striking difference in virulence, according to the infection model, may be the toxin-antitoxin (TA) systems identified so far in *P. aeruginosa*[14]. Indeed, TA systems may be involved in persistence, a bacterial tolerance strategy during antibiotics treatment[15,16]. Interestingly, we have recently reported that the latent infection model evokes some of the features of persister cells[9]. The *host inhibition of growth* (*higBA*) operon is the best studied TA system in *P. aeruginosa* so far[17]. The RNase activity of the HigB toxin is inhibited by the direct binding of the HigA antitoxin. HigA, in addition, is also able to bind to DNA and functions as a repressor, not only of *higBA* itself but also in the promoters of genes involved in virulence such as *mvfR* or type 3 and 6 secretion systems[18–20]. *higA* expression is also determined independently of that of the *higBA* operon through an internal promoter in the *higB* locus[18]. Even though the HigB toxin reduces the production of several virulence factors such as pyochelin, pyocyanin, swarming ability and biofilm formation, *higA* mutants have been reported to be more virulent than wild-type PA14 when injected into *Galleria mellonella* larvae[17,20,21].

Here, we report that *P. aeruginosa* PAO1 is not immediately virulent upon injection in the *Drosophila* host and actually fails to trigger a specific systemic innate immune response for 24 h. Virulence is acquired through a priming process within the host that can be elicited by co-exposure with N-acetyl-glucosamine (NAG). The resulting pathogenicity correlates with a switch from an ellipsoid to a bacillus shape and the ability to endure the action of a specific set of host antimicrobial peptides (AMPs). We document on the bacterial side an involvement of an interplay between *higBA*, T4P and its chemosensor PilJ, FimV, cyclic adenosine monophosphate (cAMP) signaling, and peptidoglycan synthesis for cell wall elongation that is required for the morphologic transition and switch to virulence of primed bacteria.

## Results

### The higA antitoxin mutant is susceptible to a combination of specific AMP families

Because PAO1 is hardly virulent in a recently developed oral latent infection model[9], we tested the possibility that this phenomenon might be mediated by the HigB toxin. We indeed observed a strong induction of the operon in the gut as well as a relatively modest decrease in life span induced by the Δ*higB* mutant and a more sizable one by the Δ*higBA* deletion mutant (Supplementary Fig. 1a–c). Nevertheless, the contribution of this TA system is much less important than that of melanization, a major insect host defense[9]. Unexpectedly, we noticed that the Δ*higA* mutant was less virulent than wild-type or other *higBA* mutants when ingested by melanization-deficient flies (Supplementary Fig. 1d). We therefore tested our set of mutants in a septic injury model at 18 °C, a condition akin to that used for testing melanization mutants in the latent infection model. Strikingly, the Δ*higA* mutant was much less virulent than wild-type or other mutants and displayed a decreased bacterial burden at 48 h (Fig. 1a, b). This phenotype was also obtained by directly overexpressing *higB* (Fig. 1c, d), which suggests that the Δ*higA* phenotype is caused by the unrestrained toxin, an explanation supported by the enhanced virulence of the Δ*higBA* mutant (Fig. 1a, b). Importantly, the Δ*higBA* mutant strains at 18 °C grew at the same speed as wild-type PAO1 in vitro (Supplementary Fig. 1e).

As Gram-negative bacteria infections are counteracted by the Immune Deficiency (IMD) arm of the systemic immune response, we tested whether the decreased virulence of Δ*higA* resulted from a sensitivity to the host defense. Indeed, Δ*higA* bacteria exhibited an almost fully recovered virulence in the IMD pathway mutant *kenny* (*key*) (Fig. 1e, f), a phenomenon also observed to a lesser extent in the latent infection model (compare survival of Δ*higA* in Supplementary Fig. 1d vs. f). The in vivo expression level of *higB* in Δ*higA* mutant bacteria was similar in wild-type or *key* flies (Supplementary Fig. 2a). Of note, even though the Toll pathway has been reported to participate in the host defense against *P. aeruginosa*[11], Δ*higA* bacteria did not recover their virulence in the Toll pathway mutant *MyD88* (Supplementary Fig. 2b). In keeping with a major role of the IMD pathway in restraining the virulence of Δ*higA* bacteria, we found that flies missing three categories of AMP gene families[22], *Diptericins, Drosocin/Buletin, and Attacins*, which are regulated mainly by the IMD pathway (with the exception of *Attacins* that are also co-regulated by the Toll pathway[23]), were also highly susceptible to this mutant bacterium (Fig. 1g). This was not the case for other combinations of AMP gene families still having these three AMP gene families nor for deletions of single AMP gene families tested (Fig. 1h, Supplementary Fig. 2c–g). These data suggest that the stabilization of the HigB toxin makes the bacteria vulnerable to a combination of AMPs belonging to two/three families.

### Injected P. aeruginosa is not immediately virulent nor does it trigger the systemic immune response

Having checked that Δ*higA* bacteria induce the expression of several AMP genes at least as well as wild-type PAO1 (Supplementary Fig. 3a–d), we next monitored the kinetics of IMD pathway induction using *Diptericin* steady-state transcript levels. Whereas the IMD pathway is known to be induced with a fast kinetics within 3-6 hours, we found a significant induction of *Diptericin* at 24 h but not at 12 h, as compared to a control injection of phosphate buffered saline (PBS) (Fig. 2a), in keeping with a lack of proliferation until at least 18 h (Fig. 1b). Interestingly, PGRP-LE senses small peptidoglycan (PGN) fragments released by growing and proliferating Gram-negative bacteria[24]. The finding that Δ*higA* recovers its virulence in flies missing either IMD pathway receptors PGRP-LC or PGRP-LE that senses bacterial growth further supports a lack of bacterial growth until at least 18 h (Supplementary Fig. 3e), although we cannot formally exclude an active inhibition of the IMD pathway by injected PAO1[10]. We therefore monitored when the Δ*higA* bacteria would become more susceptible to the host defense by co-injecting an equal amount of wild-type and Δ*higA* bacteria. Whereas the mixture killed the flies slightly slower than PAO1 alone (Fig. 2b), the striking observation was that the Δ*higA* bacteria grew at the same rate as wild-type bacteria until 36 h, before being killed most likely through the action of the IMD pathway by the 48 h time point (Fig. 2c). We conclude that the delayed proliferation of injected PAO1 is not caused by the activation of the IMD pathway since it becomes effective only from 36 h onwards.

### Priming in the host promotes the virulence of P. aeruginosa

We next asked whether naïve bacteria are susceptible to the action of a pre-activated IMD pathway. Whereas both wild-type and Δ*higA* bacteria displayed a similarly strongly decreased virulence when injected at 29 °C into flies that overexpress the *imd* gene at the adult stage (a strong activation of the *imd* pathway; Supplementary Fig. 4a), the decrease of virulence at 18 °C was less pronounced in flies that had been pre-challenged by the injection of heat-killed PAO1 24 hours earlier, with bacterial proliferation observed at 48 but not 24 h (Supplementary Fig. 4b, c). These data taken together with those described in the paragraph above imply that wild-type PAO1 injected into wild-type naïve flies become resistant to the action of the IMD pathway after some 24-36 hours. Next, we pre-challenged wild-type flies by the injection of some 100 green fluorescent protein (GFP)-labeled PAO1 followed by the

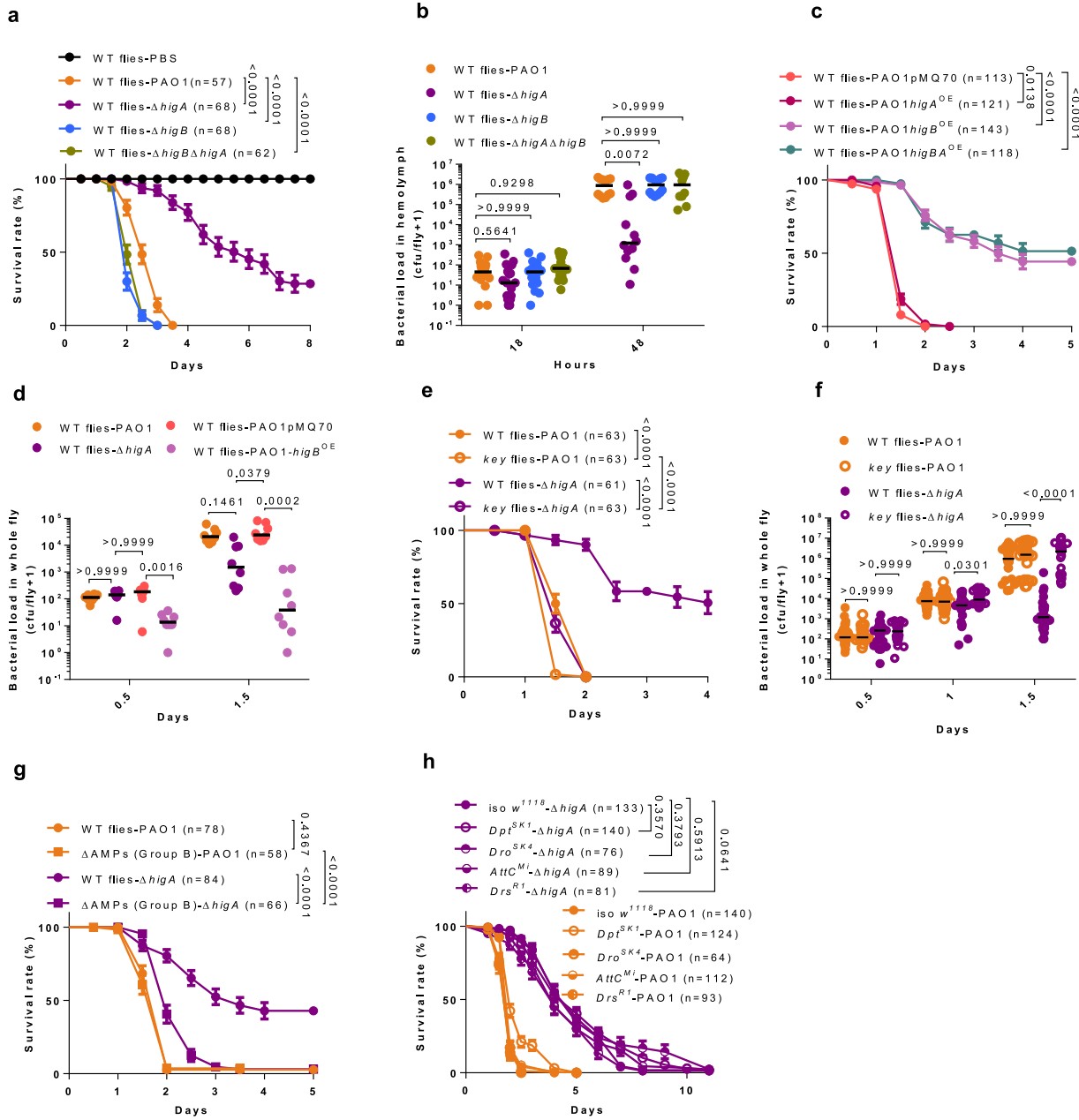

**Fig. 1 | Type II toxin HigB enhances _Pseudomonas aeruginosa_ sensitivity to the combined action of three AMPs in _Drosophila_. a** Survival of wild-type flies injected with a set of _higBA_ mutants in PAO1. **b** Bacterial load corresponding to (**a**). **c** Survival of wild-type flies injected with a set of _higBA_ overexpression strains. **d** Bacterial load corresponding to (**c**). **e** Survival of wild-type and _key_ flies injected with Δ_higA_. **f** Bacterial load corresponding to (**e**). **g** Survival of Group B AMPs-deficient flies injected with Δ_higA_. Group B indicates flies lacking _Diptericins_, _Drosocin/Buletin_, and _Attacins_. **h** Survival of single AMP gene-deficient flies injected with Δ_higA_. Bars represent the standard error of the mean (**a, c, e**) or the median (**b, d, f**). Fly and bacteria strains are denoted by distinct symbol shapes (filled circles

represent wild-type flies, unfilled circles indicate _key_ flies, filled squares denote Group B AMP-deficient flies, variant circle shapes (_e.g._, half-filled) represent individual AMP-deficient strains) and colors (wild type PAO1, Δ_higA_, Δ_higB_, Δ_higB_-Δ_higB_, PAO1_higA_^OE, PAO1_higB_^OE, PAO1_higBA_^OE were distinguished in turn by orange, dark purple, light blue, yellowish-green, dark red, light purple and indigo) respectively (hereinafter the same). Experiments were repeated three times (**a,c, e–h**; pooled data; $n = 24$ in **f**), twice (**b**; pooled data; $n = 18$ at 18hpi, $n = 12$ at 48hpi), once (**d**, $n = 8$). Statistical analysis was done by Logrank (Mantel-Cox test) in (**a, c,e, g, h**), by Kruskal-Wallis test with Dunn's post-hoc test in (**b, d, f**). Source data are provided as a Source Data file.

injection of some 1000 red fluorescent protein (RFP)-labeled PAO1 24 h thereafter, a quantity similar to that measured in flies 24 h after injection (Fig. 2d; corresponding survival experiment shown in Supplementary Fig. 4d). Strikingly, the titer of RFP bacteria was significantly reduced within 1 h (25 h time point) and those bacteria were cleared by 6 h (30 h time point) by wild-type but not _key_ flies (Fig. 2e), suggesting that GFP-PAO1 had adapted to the host and its immune response, which was not the case of the naïve RFP-bacteria. Next, we retrieved bacteria from an

injected host and injected them into flies pre-challenged with heat-killed PAO1 24 h earlier. Remarkably, the primed bacteria were immediately virulent as compared to naïve bacteria, whether the bacteria had been retrieved from wild-type or _key_ flies (Fig. 2f; note that the absolute number of injected bacteria does not have a strong effect on survival curves (Supplementary Fig. 4e–f)). This experiment further establishes that priming is not initiated by an exposure to the IMD immune response. Moreover, retrieved bacteria were also more virulent when

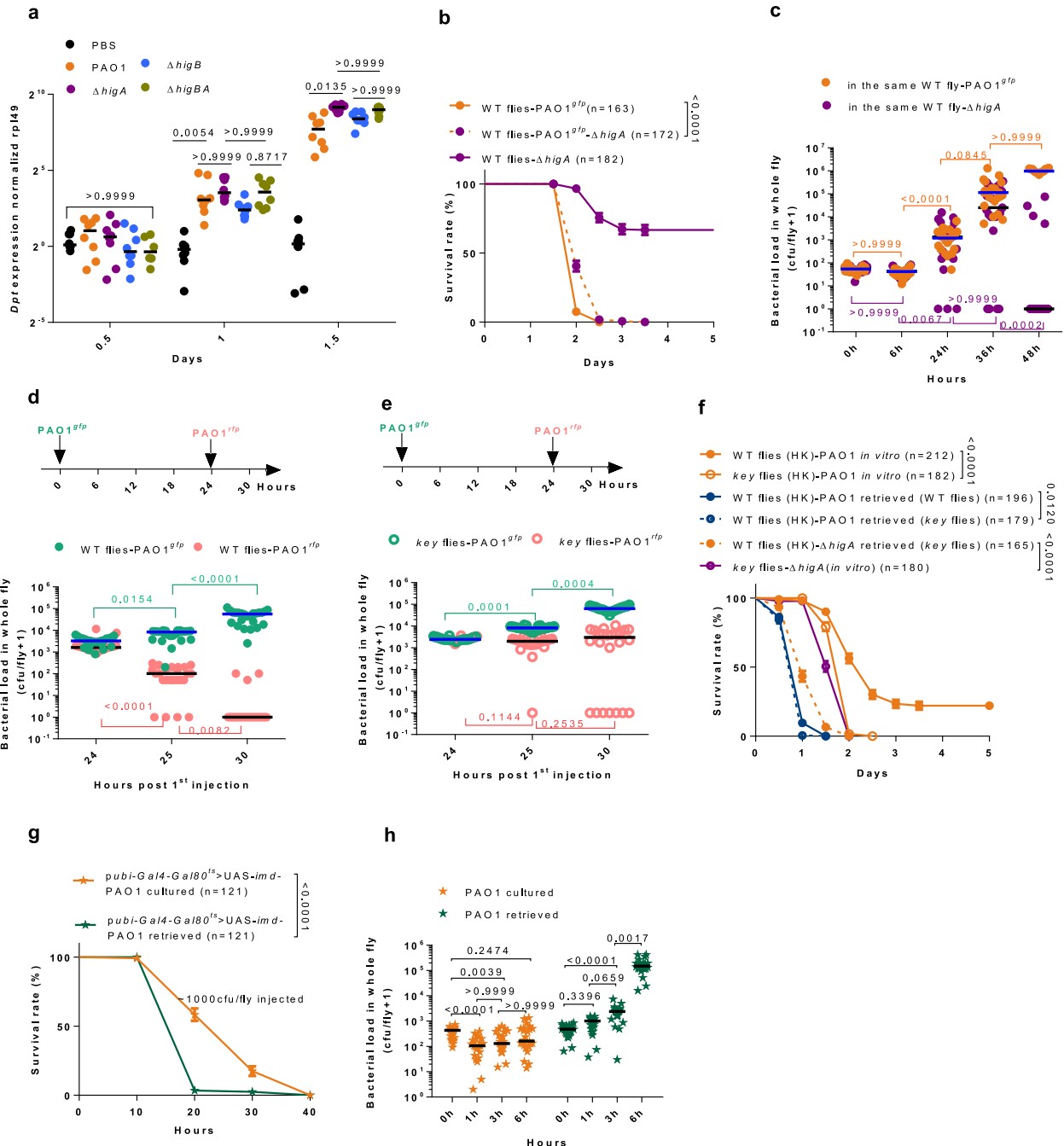

**Fig. 2 | Wild-type *P. aeruginosa* eludes action of antimicrobial peptides by priming. a** Kinetics of *Dpt* steady-state transcript levels after *P. aeruginosa* injection. **b** Co-injection of PAO1 and Δ*higA* in wild-type flies. **c** Bacterial load corresponding to (**b**). **d, e** Bacterial load in wild-type flies (**d**) and *key* flies (**e**) first injected with PAO1*gfp* and then secondarily injected with PAO1*rfp* 24 h later. Survival data are shown in Supplementary Fig. 4d. **f** Virulence of retrieved bacteria. The fly genotypes enclosed by parentheses indicate the donors for retrieved bacteria. HK enclosed by parentheses refers to flies pre-challenged by heat-killed bacteria. **g** Survival of *imd*-overexpressing flies challenged with naive or retrieved bacteria. **h** Bacterial load

corresponding to (**g**). Bars represent the median (**a, c, d, e, h**) or the standard error of the mean (**b, f, g**). The GFP and RFP labeled bacteria are denoted by light green and pink, respectively. PAO1 injected into *imd*^OE^ flies were denoted by stars and distinguished by orange (PAO1 cultured) and ultramarine (PAO1 retrieved). Experiments were repeated three times (**a–h**; pooled data; *n* = 8 in a; *n* = 24 in **c, d, e, h**). Statistical analysis was done by Logrank (Mantel-Cox test) in (**b, f–g**), by Kruskal-Wallis test with Dunn's post-hoc test in (**a, c–e, h**). Source data are provided as a Source Data file.

injected in *imd*-overexpressing flies (Fig. 2g). Importantly, the retrieved bacteria started to proliferate shortly after their injection in wild-type flies (Fig. 2h). Moreover, the Δ*higA* retrieved from wild type flies at 24hpi could not kill naive flies as fast as those retrieved from *key* flies (Supplementary Fig. 4g in contrast to Fig. 2f).

We conclude that priming first allows the bacteria to proliferate actively and second allows them to withstand the action of the IMD

pathway, likely by resisting the combined antimicrobial action of the two/three AMP families.

## Priming correlates with a morphological switch
We had noted in an earlier study that PAO1 harbor an ellipsoid shape when grown in vitro (in Luria-Bertani (LB) or brain-heart infusion (BHI) medium) and that bacteria retrieved initially from the hemolymph in

the latent infection model display instead a bacillus conformation[9]. PAO1 also adopts a rod shape in wild-type or *key* hosts 48 hours after injection. Interestingly, we found that Δ*higA* mutants displayed a rounded morphology in wild-type and a rod one in *key* flies (Fig. 3a-b'). The transition from ellipsoid to rod occurs during the priming period: whereas bacteria still present mostly an ellipsoid morphology at 13 h, a mixture of both forms was observed 16 h after injection whereas mostly rods were observed 24 hours after infection, a time at which bacteria start proliferating and have been primed (Fig. 3c-c', Supplementary Fig. 5a-a'). Thus, virulence correlates with a bacterial morphology transition.

## N-acetyl-glucosamine promotes the priming of *P. aeruginosa* and a morphological switch

Since priming does not depend on exposure to the IMD host defense, we reasoned that it might result from an attack by enzymes circulating in the hemolymph such as bacterial or host lysozymes or chitinases. Pre-exposure of bacteria to lysozyme in vitro for 30 minutes in the presence or absence of Ethylenediamine tetraacetic acid (EDTA) was sufficient to increase the virulence of either wild-type or Δ*higA* bacteria injected into wild-type or *key* flies (Fig. 3d, Supplementary Fig. 5b). The pre-injection of a chitinase into flies 30 minutes prior to a challenge with PAO1 similarly led to enhanced virulence of the injected bacteria (Fig. 3e). Of note, *P. aeruginosa* also encodes a chitinase gene that is however not required for its virulence in PA14 bacteria and which is actually down-regulated when PAO1 bacteria were injected into wild-type or *key* flies (Supplementary Fig. 5c, d)[25]. The hydrolase activity of both enzymes releases N-acetyl-glucosamine (NAG). We therefore co-injected PAO1 and NAG and observed again an enhanced virulence (Fig. 3f), which correlates with an earlier transition to rod morphology as most bacteria exhibited it 13 h post co-injection (Supplementary Fig. 5a-a'). As expected from a switch occurring earlier, the injection of NAG 24 h after the initial PAO1 challenge did not alter the virulence of PAO1 (Supplementary Fig. 5e).

An alternate possibility is that NAG is released directly through the metabolism of PGN. In such case, the digestion of PGN will initially yield both NAG and N-acetyl muramic acid (NAM). When we co-injected PAO1 and an equimolar mixture of NAG and NAM, we detected a decreased virulence instead (Fig. 3g), suggesting that PGN hydrolysis may not mediate priming. Unexpectedly, the co-injection of PAO1 and NAM alone almost abolished the virulence of the bacteria in wild-type hosts and severely decreased it in *key* flies (Fig. 3h); this striking result further supports the virulence-promoting properties of NAG (compare Fig. 3h–g). Again, an injection of NAM 24 h after the initial PAO1 challenge had no impact on the course of the infection (Supplementary Fig. 5f), in keeping with a lack of morphological transition 24 h after the co-injection of PAO1 and NAM, which might however, start at 48 h (Supplementary Fig. 5a-a). Of note, the culture of PAO1 with either NAM or NAG did not alter much the growth of PAO1 in vitro, and in any case, was clearly distinct from in vivo growth (Supplementary Fig. 5g) not did it impact the virulence of the purified injected bacteria (Supplementary Fig. 5h-i). NAM and NAG did also not inhibit the lysozyme antimicrobial activity against *Micrococcus luteus* in vitro (Supplementary Fig. 5j).

We conclude that NAG likely originating from the host promotes the transition to a bacillus morphology, which correlates with virulence.

## A proteomic analysis reveals an involvement of the type IV pili system in mediating the effects of the higBA operon

Whereas Δ*higA* bacteria retrieved from *key* mutants rapidly kill wild-type flies pre-challenged with heat-killed PAO1 within a day and a half (Fig. 2f), these bacteria retrieved from wild-type hosts and injected in wild-type flies killed them only slightly faster than naïve Δ*higA* bacteria (Supplementary Fig. 4g). This observation suggests that Δ*higA* bacteria may be poorly primed when exposed to the action of the IMD pathway,

in keeping with their ellipsoid shape. To better understand the role of the *higBA* operon, we assessed the quantitative expression of PAO1 proteins in Δ*higA*, Δ*higB*, and Δ*higBA* bacteria relative to those found in wild-type bacteria cultured in vitro (Fig. 4a, Supplementary Fig. 6a and Supplementary Data 1). Proteins belonging to the type three secretion system (T3SS) and to the phenazine biosynthesis operon were significantly repressed in Δ*higA* mutants and respectively appeared to be either somewhat induced or unaltered in Δ*higB* or Δ*higBA* mutants. Thus, the HigA antitoxin appears to promote the expression of the T3SS. The ectopic expression of *exsA*, which encodes a major regulator of T3SS operon expression[26], did not rescue the lessened virulence phenotype of injected Δ*higA*, suggesting that the T3SS may not be a major contributor to the susceptibility of Δ*higA* mutants to the IMD response (Supplementary Fig. 6b)[12]. Indeed, Δ*higB* and Δ*higBA* are equally virulent in the injection model despite the induction of the T3SS in the former and not the latter (Fig. 1a). Likewise, phenazines are also unlikely to contribute to pathogenesis in the septic injury model since Δ*higB* and Δ*higBA* mutants are more virulent than wild-type bacteria in the injection model despite their decreased expression of phenazine genes observed in vitro (Fig. 1a).

In contrast, most genes required for the formation of type IV pili (T4P) are expressed in Δ*higB* and Δ*higBA* mutants at levels similar to those encountered in wild-type bacteria but are clearly repressed in the Δ*higA* mutants, especially the *pilQ* and *fimV* gene which respectively encode the type IV pilin secretin channel embedded in the outer membrane of the bacteria and a PGN-binding protein required for assembly of the secretin channel[27]. Unexpectedly, the decreased protein expression of T4P components was not mirrored at the transcript level (Supplementary Fig. 6c), even though the HigB toxin is thought to destabilize mRNAs[28–30].

The T4P is required for twitching motility and *higA* is also required for twitching motility (Supplementary Fig. 6d), further supporting a link between the *higBA* operon and the T4P. *pilJ* encodes the chemosensor of the Pil/Chp chemosensing system[4,31]. D'Argenio et al. showed that it is chemo-sensation and not twitching motility per se that is important for *P. aeruginosa* virulence in *Drosophila* since 24 mutants affecting twitching motility still displayed a normal virulence in flies[32]. We confirmed that the Δ*pilJ* mutant was less virulent than wild-type PAO1 (Fig. 4b). However, whereas D'Argenio et al.[32] had not observed a difference in the bacterial burden between wild-type and mutants affecting the *pilGHIJKL chpABCDE* operon at the time points examined, up to 18 hours, we observed a progressively decreasing Δ*pilJ* bacterial load from 24 hours onward in the collected hemolymph and from 36 hours onward in whole flies. These observations suggest that like Δ*higA* mutants, Δ*pilJ* mutants are sensitive to the fly's immune response (Supplementary Fig. 6e, f). Indeed, Δ*pilJ* mutants recovered full virulence when injected into *key* and group B mutants (Fig. 4b, Supplementary Fig. 6g). We next overexpressed *pilJ* in a Δ*higA* mutant background and significantly rescued the lessened virulence Δ*higA* phenotype in wild-type flies (Fig. 4c). However, the virulence of the *pilJ*-rescued Δ*higA* bacteria was somewhat reduced relative to that of Δ*higA* bacteria in *key* mutant flies, revealing a modestly impaired fitness of these bacteria, at least in vivo.

## The cAMP-Vfr pathway but not the T4P itself is important for *P. aeruginosa* virulence

PilJ, together with other Chp proteins, activates a signaling pathway that regulates the activity of the CyaB cAMP cyclase. cAMP levels control bacterial cell morphology and also activate the Vfr transcription factor, which regulates the expression of several virulence genes[4,31,33–36]. The overexpression of the adenylate-cyclase coding gene *cyaB* partially rescued the impaired virulence of Δ*higA* mutants, even though this overexpression slightly impairs the fitness of these bacteria (in *key* flies) (Fig. 4d), as was the case for *pilJ* overexpression. cAMP levels activate the Vfr transcription factor. Interestingly, we

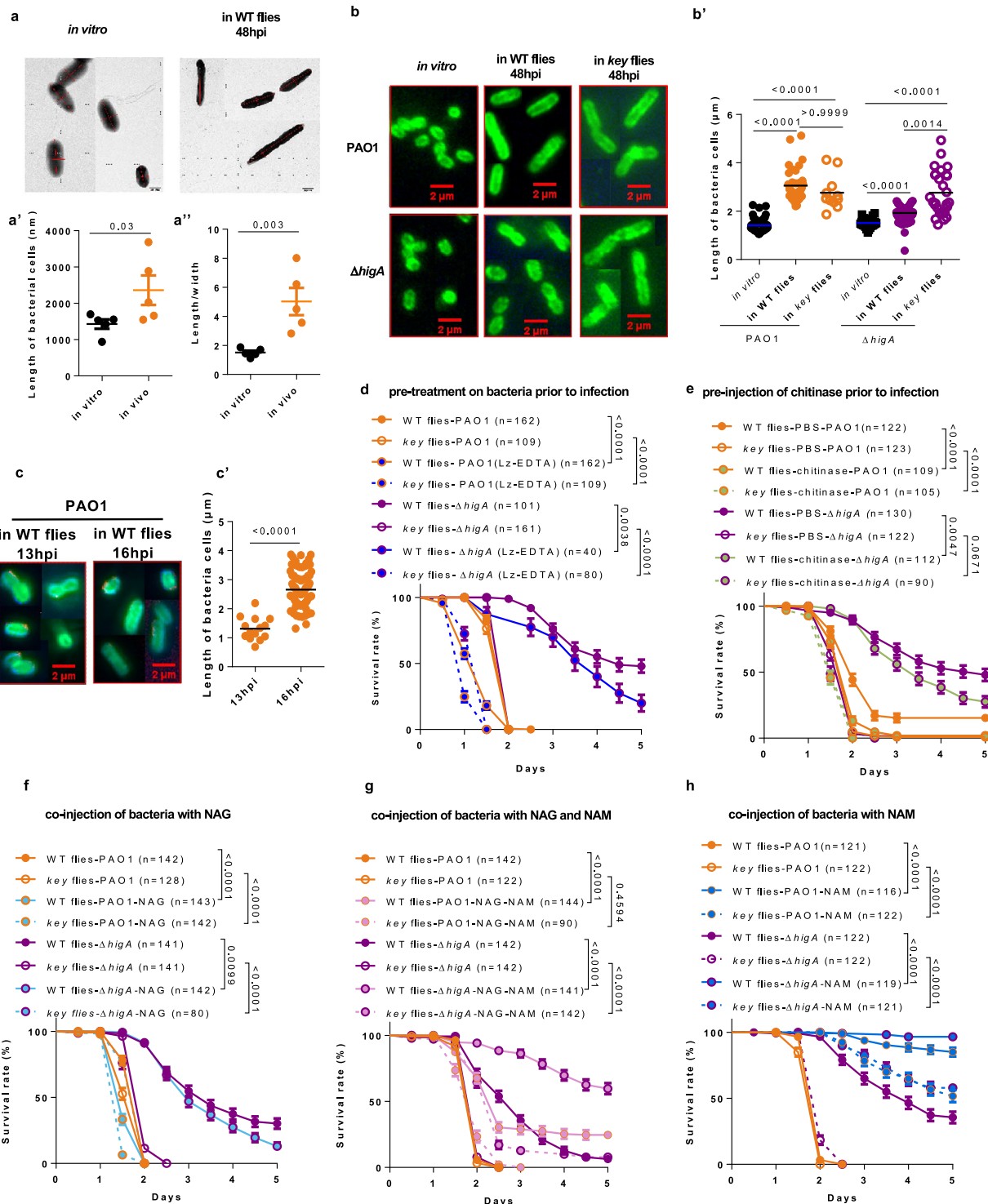

**Fig. 3 | A morphology switch occurs during *P. aeruginosa* priming in vivo. a–c** Characteristic morphology of *P. aeruginosa* PAO1 and Δ*higA* in vitro, in wild-type or *key* flies. Because of the limited number of bacterial cells in most visual fields, all figures in vivo are composite assemblies from different visual fields. Morphology of PAO1 was visualized by transmission electron microscopy in vitro and in vivo (**a**). Morphology of PAO1 and Δ*higA* were visualized by O5 antibody staining under fluorescence microscopy in vitro, in tissues of wild-type or *key* flies in (**b**) and at different time points in (**c**). The lengths of bacteria and the length/width ratios of bacteria were quantified respectively in (**a', b', c'**) and (**a''**). **d** Survival of flies infected by lysozyme treated *P. aeruginosa*. EDTA, ethylene diamine tetraacetic acid; Lz, lysozyme. **e** Survival of flies injected first with chitinase and then with *P. aeruginosa* 30 min later. **f–h** Survival of flies co-injected with *P. aeruginosa* in a N-acetyl-glucosamine (NAG)-containing solution (**f**), NAG- and N-acetyl-muramic acid (NAM)-containing solution (**g**) and NAM-containing solution (**h**). Bars represent the mean (**a, a', b', c'**) or the standard error of the mean (**d–h**). The treatments of bacteria are denoted by distinct colors (Lz treatment: bright blue; chitinase treatment: yellowish-blue; NAG supplementation: light blue; NAG and NAM supplemented: pink; NAM supplemented: sky blue; hereinafter the same color code is used). Experiments were repeated three times (**b–h**; pooled data; For panel **b'**, the following data points were analyzed: (i) PAO1 in vitro (*n* = 64), WT flies (*n* = 36), and *key* flies (*n* = 11); (ii) Δ*higA* in vitro (*n* = 25), WT flies (*n* = 65), and *key* flies (*n* = 27). For panel **c'**, the following data points were analyzed: 13hpi (*n* = 14), 16hpi (*n* = 86)), once (**a-a''**; *n* = 5). Statistical analysis was done by one tailed Student's t-test in (**a'-a''**, **c'**), by Kruskal-Wallis test with Dunn's post-hoc test in (**b'**), by Logrank (Mantel-Cox test) in (**d–h**). Source data are provided as a Source Data file.

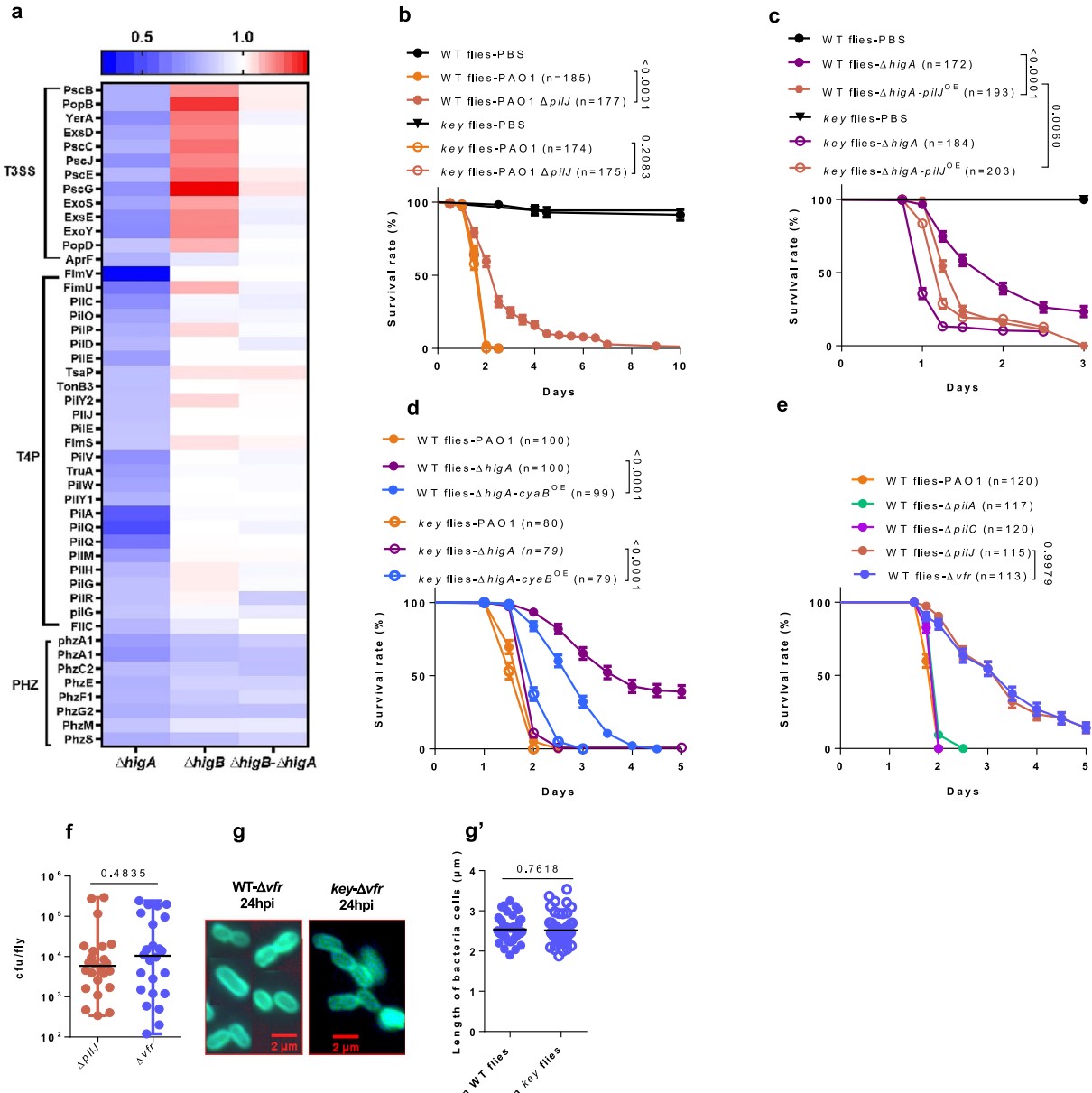

**Fig. 4 | A type IV pili chemosensor PilJ-cAMP-Vfr axis is required for virulence and priming. a** Heatmap of differentially expressed proteins in mutants affecting *higBA* based on in vitro proteomic analysis. **b** Survival of flies challenged by Δ*pilJ*. **c, d** Survival of flies challenged by Δ*higA* bacteria harboring a plasmid overexpressing *pilJ* (**c**) or *cyaB* (**d**). **e** Survival of flies challenged by different mutants related to the T4P, including Δ*pilJ*, Δ*pilA*, Δ*pilC*, Δ*vfr*. **f** Bacterial load corresponding to (**e**). **g** Morphology of Δ*vfr* in the hemolymph wild-type and *key* flies visualized by O5 antibody staining under fluorescence microscope. Because of the limited number of bacterial cells in most visual fields, all figures are composites from different visual fields. **g'** Quantitative analysis of cell length in (**g**). Bars represent the standard error of the mean (**b–e**), the median (**f**), or the mean (**g'**). Bacteria strains are denoted by distinct colors (Δ*pilJ* and Δ*higA-pilJ*^OE: brick red; Δ*higA-cyaB*^OE: sky blue; Δ*vfr*: blue-gray; hereinafter the same color code is used). Experiments were repeated three times (**b-g'**; pooled data; **f** (*n* = 24); **g'**: in WT flies (*n* = 51), in *key* flies (*n* = 50)); once with three replicates (**a**). Statistical analysis was done by Logrank (Mantel-Cox test) in (**b–e**), by two tailed Student's t-test in (**f, g'**). n.s.: not significant. Source data are provided as a Source Data file.

found *vfr* expression to be induced in vivo some 40 h after injection (Supplementary Fig. 7a). To further assess the role of Vfr, we generated a deletion mutant and found it to display a reduced virulence equivalent to that exhibited by Δ*pilJ* mutants with respect to both survival and bacterial load (Fig. 4e, f). However, its overexpression did not appear to rescue the Δ*pilJ* mutant phenotype. Conversely, *pilJ* overexpression also did not rescue the Δ*vfr* mutant phenotype (Supplementary Fig. 7b). The morphology of Δ*vfr* mutants resembled that of Δ*higA* mutant injected into wild-type flies (Fig. 4g-g), albeit in contrast to the latter (Fig. 3b-b), it did not become rod-like in *key* mutants. This may perhaps reflect a requirement for Vfr function in lateral cell wall elongation (there is still some PilJ made in Δ*higA* mutants).

The T4P is involved in surface sensing and retains the information that it has been in contact with a surface over several bacterial generations[37–42]. We therefore asked whether exposure of PAO1 to a surface in vitro would mimic the effect of priming as injected bacteria may temporarily become associated to the surface of host tissues. Even though the contact with 1.5% agarose did induce the expression of the *xphA* gene (Supplementary Fig. 7c), this gene did not appear to be induced in vivo (Supplementary Fig. 7d). Importantly, bacteria that had been pre-exposed to this surface did not display an enhanced virulence when compared to naïve *in vitro*-grown bacteria (Supplementary Fig. 7e). Thus, priming is not related to the sensing of surfaces, as further underscored by the near normal virulence of mutant

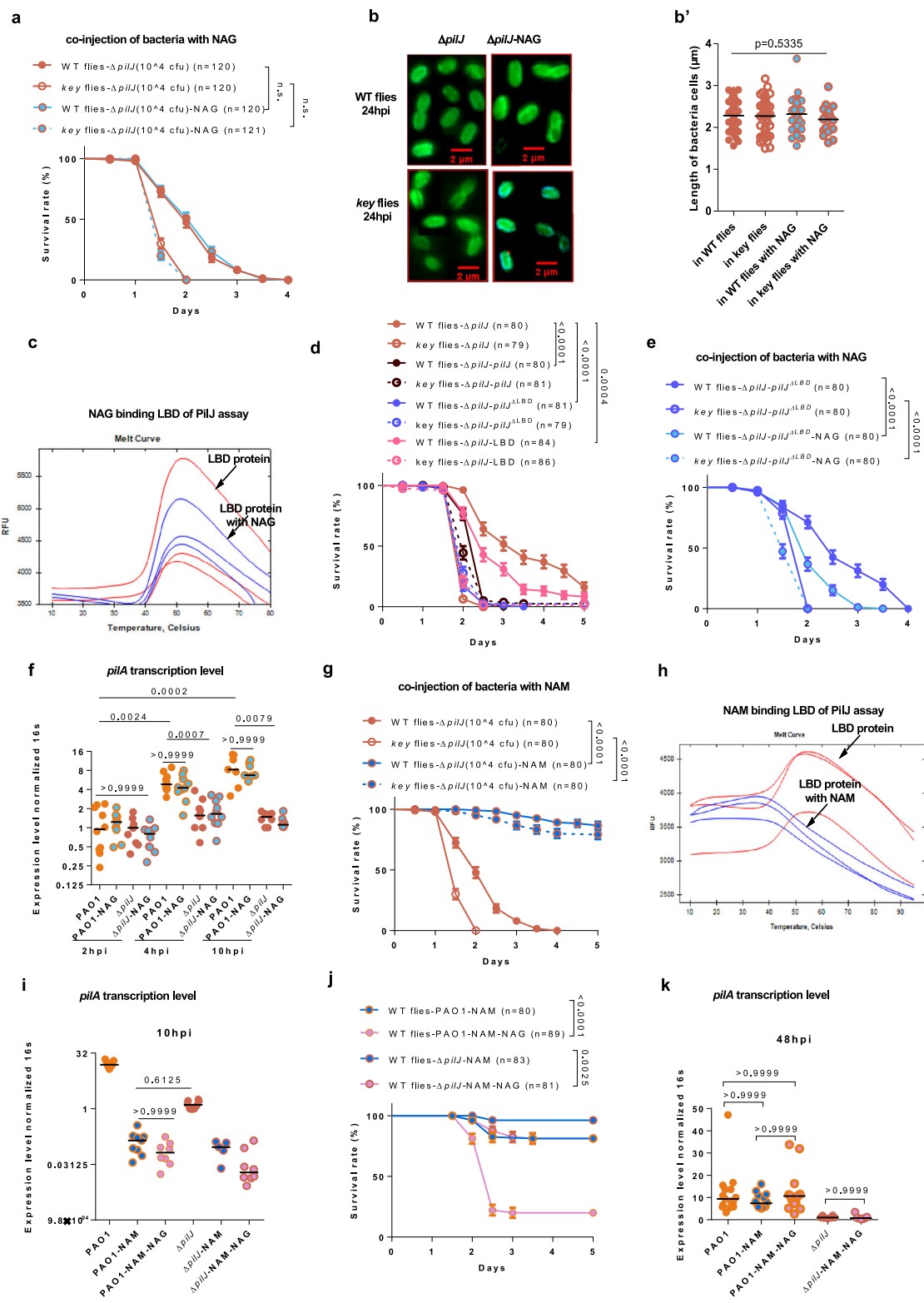

bacteria with a defective T4P (Δ*pilA*, Δ*pilC*), as reported previously[32] (Fig. 4e and Supplementary Fig. 7f).

## The PilJ chemosensor is required for the priming of P. aeruginosa by NAG

Whereas the virulence of naïve wild-type bacteria is enhanced close to the level of primed bacteria upon co-injection with NAG into wild-type flies (Fig. 3f), we did not detect any change in the survival rate of wild-type flies injected with Δ*pilJ* bacteria in the presence or absence of NAG

(Fig. 5a). Of note, the virulence of Δ*higA* bacteria was still somewhat increased when co-injected with NAG (Fig. 3f), in keeping with a decreased but not abolished expression of PilJ in these mutants (Fig. 4a). The shape of Δ*pilJ* bacteria remained ellipsoid 20 hours after infection of either wild-type or *key* flies. The co-injection of NAG also did not alter the morphology of the injected Δ*pilJ* bacteria (Fig. 5b-b).

Previously, PilJ has been documented to be required for sensing phenol soluble modulins emitted by *Staphylococcus aureus*, albeit direct binding of these peptides to PilJ has not been yet documented[43].

**Fig. 5 | N-acetyl-muramic acid but not N-acetyl-glucosamine exerts potent modulation of the cAMP-Vfr signaling. a** Survival of flies injected with Δ*pilJ* in NAG-containing solution. **b** Characteristic morphology of Δ*pilJ* bacteria with NAG co-injection in the tissues of wild-type and *key* flies visualized by O5 antibody staining under fluorescence microscope. Because of the limited number of bacterial cells in most visual fields, all figures are composites from different visual fields. **b'** Quantitative analysis of cell length in (**b**). **c** Ligand binding assay was determined by thermal shift assay with NAG (blue lines) using protein without ligand as control (red lines). **d** Survival of flies challenged by Δ*pilJ* harboring a plasmid overexpressing full-length *pilJ*, *pilJ*^ΔLBD, or LBD alone. **e** Survival of flies injected with Δ*pilJ* harboring a plasmid overexpressing *pilJ*^ΔLBD in NAG-containing solution. **f** Kinetics of transcript levels of the readout gene *pilA* for cAMP-Vfr signaling in wild-type *P. aeruginosa* and Δ*pilJ* upon NAG in vivo. **g** Survival of flies challenged with Δ*pilJ* in NAM-containing solution. **h** Ligand binding assay was

determined by thermal shift assay with NAM (blue lines) using protein without ligand as control (red lines). **i** Transcript levels of the readout gene *pilA* for cAMP-Vfr signaling in wild-type *P. aeruginosa* and Δ*pilJ* upon NAM 10 hours post infection. **j** Survival of flies injected with wild-type *P. aeruginosa* and Δ*pilJ* in NAM or NAM-NAG-containing solution. **k** Transcript levels of the readout gene *pilA* for cAMP-Vfr signaling in wild-type *P. aeruginosa* and Δ*pilJ* upon NAM 48 h post infection. Bars represent the standard error of the mean (**a, d-e, j**), the mean (**c'**), or the median (**f, i, k**). Experiments were repeated three times (**a-b', d-g, i-k**; pooled data; **b'**: WT flies (*n* = 64), *key* flies (*n* = 67), WT flies with NAG (*n* = 50), *key* flies with NAG (*n* = 20); **k**: PAO1 groups (*n* = 15), Δ*pilJ* groups (*n* = 8)), twice in (**f,i**; pooled data; *n* = 7-9), while one representative result of three independent experiments with consistent findings was displayed in (**c, h**). Statistical analysis was done by Logrank (Mantel-Cox test) in (**a, d, e, g, j**), by Kruskal-Wallis test with Dunn's post-hoc test in (**b', f, i, k**). n.s.: not significant. Source data are provided as a Source Data file.

A thermal shift assay has been employed to demonstrate a direct binding of the PilJ domain of the McpN chemoreceptor to nitrate[44]. To directly test the hypothesis of a direct binding of PilJ to NAG, we expressed either recombinant full length PilJ or a fragment containing the tandem ligand-binding domains (LBD)[45] (Supplementary Fig. 8a, b). As the former is an integral membrane protein, it was found to be unstable when the temperature was increased to 32 °C and its behavior was not changed by co-incubation with either NAG or NAM (Supplementary Fig. 8c). As shown in Fig. 5c, the addition of NAG did not influence the thermal shift of the LBD domains. Finally, we used a genetic approach to assess the importance of the LBD domains of PilJ. The overexpression of full-length *pilJ* rescued the Δ*pilJ* mutant phenotype (Fig. 5d). The overexpression of the sole PilJ LBD domains mildly but nevertheless significantly rescued the virulence of Δ*pilJ* mutants. Unexpectedly, the overexpression of a *pilJ* gene lacking the sequences coding the tandem LBD domains was able to fully rescue the virulence of Δ*pilJ* mutants, suggesting that the LBD domains are not essential for the function of PilJ in the priming process (Fig. 5d). Thus, NAG may interact directly or indirectly through other domains of the PilJ protein, which does not function as a canonical chemosensor of this metabolite. To confirm this finding, we co-injected in wild-type and *key* flies the *pilJ* mutants carrying a plasmid that overexpresses the truncated *pilJ* gene lacking the tandem LBD coding sequences and NAG. This treatment restored virulence (Fig. 5e), in contrast to the Δ*pilJ* mutants exposed to NAG in vivo (Fig. 5a). We next measured the transcript level of *pilA*, a key readout gene for cAMP-Vfr signaling[46], under these different in vivo conditions. Whereas the cAMP-Vfr signaling pathway was activated in wild-type PAO1 by 4 hours post-injection, it remained inactive in *pilJ* mutants (Fig. 5f). Of note, NAG supplementation did not influence the induction of *pilA* expression, suggesting that NAG may influence a PilJ-dependent mechanism that is distinct from the activation of the cAMP-Vfr pathway.

In contrast, the co-injection of NAM still potently decreased the virulence of *pilJ* mutants (Fig. 5g). Unfortunately, the co-incubation of the LBD domains with NAM led to the denaturation of the LBD domains (Fig. 5h), foreclosing any interpretation.

Interestingly, the NAM treatment, with or without the addition of NAG, prevented the activation of the cAMP-Vfr pathway in both PAO1 and *pilJ* mutants, as monitored by *pilA* transcript levels 10 h after injection that were expressed only at very low levels (Fig. 5i). Nevertheless, the co-injection of NAG with NAM did rescue the virulence of PAO1 after 48 h (Fig. 5j; Supplementary Fig 8d). By monitoring *pilA* expression at 48 h post-challenge, we found that the cAMP-Vfr pathway did ultimately get activated despite the presence of NAM, in a *pilJ*-dependent manner (Fig. 5k). We conclude that NAM treatment inhibits or prevents the activation of the cAMP-Vfr pathway during the early stages of the infection, when priming occurs, by an as yet uncharacterized mechanism. While NAM treatment alone prevents a switch to virulence for at least five days (Fig. 5g), it is, however, able to counteract the effect of NAG on PilJ only for a period limited to the early

stages of the infection. Altogether, these data suggest that NAG promotes the switch to virulence independently of the activation of the cAMP-Vfr pathway.

## *fimV* functions downstream of *higBA*

As noted earlier, FimV levels are strongly reduced in Δ*higA*. The FimV PGN-binding protein is associated with the T4P and *fimV* mutants are more susceptible to injected PAO1[32]. In contrast to *vfr* (Supplementary Fig. 9a), the overexpression of *fimV* did markedly rescue the Δ*higA* mutant phenotype, actually better than *cyaB* overexpression (Fig. 4d), even though the *higA* bacteria overexpressing *fimV* displayed a somewhat reduced fitness as observed in vivo when injected into *key* mutants (Fig. 6a). *fimV* has been reported to be genetically upstream of *cyaB*; thus, the stronger intensity of its rescue of Δ*higA*, as compared to *cyaB* overexpression, may mirror a *cyaB*-independent function of *fimV*, in keeping with its proposed cAMP-independent function[47]. However, *fimV* overexpression failed to rescue either the Δ*pilJ* or the Δ*vfr* mutant phenotypes (Supplementary Fig. 9b) whereas it did rescue the Δ*higA* phenotype (Fig. 6a). This may reflect again the fact that there is still some PilJ made in the Δ*higA* mutants and that the cAMP pathway may still be partially activated in these mutants, as displayed schematically in Supplementary Fig. 9c.

## The virulence of PAO1 depends on FimV polar localization

The restored virulence of Δ*higA* by *fimV* overexpression correlated with a bacillus shape (Fig. 6b-b'). As FimV plays a role in positioning the T4P to the site of future cell division[48], which will become the poles of the two new bacterial cells, we investigated its subcellular location within the bacterial cell by overexpressing a GFP-tagged form, which is functional since bacteria were more elongated (mean of 4 μm) than wild-type PAO1 (mean of 3 μm; compare Fig. 3b-b' to Fig. 6c, see below). Strikingly, upon its overexpression, whereas tagged FimV was found distributed rather evenly throughout the cell internal surface in 60% of cells grown in BHB for 16 hours, it was found to be enriched at the poles of all bacteria injected into wild-type flies 24 h after challenge (Fig. 6c), which suggests that bacterial pole organization changes as a result of priming. Actually, these bacteria were homogenously longer in vivo than in vitro, for which size varied extensively (Fig. 6c, see below). The FimV protein contains several identified domains, including a LysM, a coiled-coil (CC), and a cytoplasmic domain (CD) (Fig. 6d)[27]. The overexpression of constructs encoding truncated FimV proteins in Δ*higA* mutants revealed that the LysM domain is largely dispensable for the rescue, in contrast to the CC and CD domains (Fig. 6e). The expression of GFP-tagged FimV missing either the CC (N-terminal tag) or CD (C-terminal tag) domains revealed that there is still an enrichment at the polar poles when the CD domain is missing but a very limited one when the CC domain is missing (Fig. 6f, Supplementary Fig. 9d), suggesting that the latter is required for polar localization of FimV. Indeed, the morphology of bacteria overexpressing the CC deleted *fimV* construct remained ellipsoid (Supplementary Fig. 9d-d'').

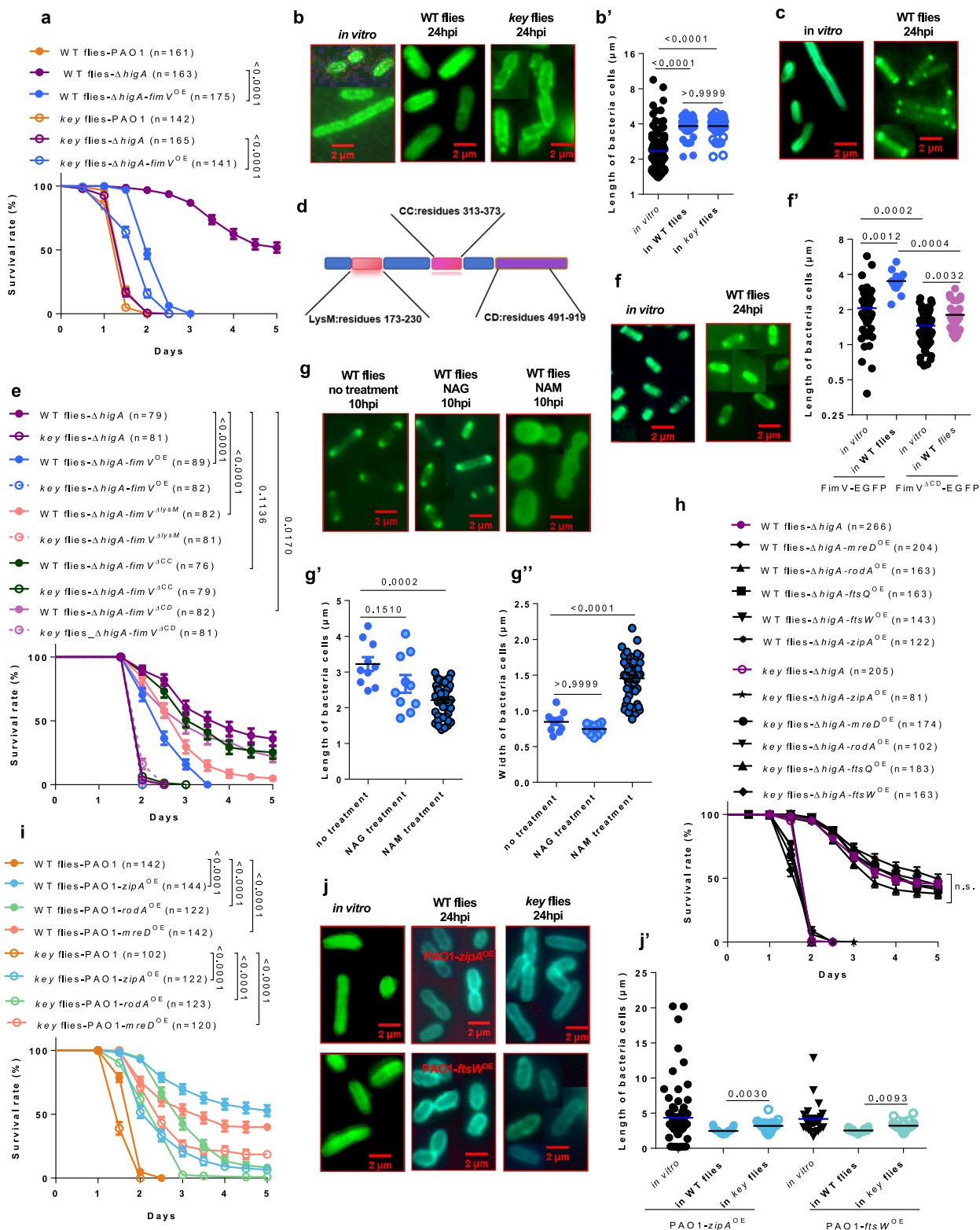

The CD domain is solely required for extensive lateral cell wall peptidoglycan synthesis since the bacteria are shorter than when overexpressing full *fimV* yet retain a rod shape (Fig. 6f', Supplementary Fig. 9d-d'''). Of note, a polar distribution of FimV is also likely required for the elongation of the bacteria as the bacteria overexpressing FimV lacking the CC domain appeared to be more roundish than elongated (Supplementary Fig. 9d-d'''). The overexpression of the CC or CD deletion mutant genes impaired the virulence of the overexpressing bacteria when injected into wild-type flies, albeit the tagged deletion

protein mutants were somewhat deleterious as the virulence of the corresponding bacteria in *key* mutants was mildly impaired (Supplementary Fig. 9e).

Since both NAM and NAG influence the virulence of injected PAO1 in a cAMP/Vfr independent manner, we next asked whether they might impact the localization of FimV to the poles of bacteria. We first investigated this issue in *fimV*-overexpressing bacteria cultured in vitro. The exposure to NAG significantly increased the length of bacteria as monitored six hours after culture (Supplementary

**Fig. 6 | FimV sub-cellular placement determines bacterial polarity and morphology as well as virulence in vivo. a** Survival of flies challenged by Δ*higA* bacteria overexpressing *fimV*. **b** Characteristic morphology of Δ*higA* bacteria overexpressing *fimV* in vitro and in vivo in wild-type or *key* tissues visualized by O5 antibody staining under fluorescence microscope. **b′** Quantitative analysis of cell length. **c** Characteristic morphology of bacteria overexpressing FimV fused EGFP and the location of FimV-EGFP in *P. aeruginosa* in vitro and in tissues of wild-type flies. **d** Scheme of different domains of FimV protein. **e** Survival of flies challenged with Δ*higA* harboring plasmids overexpressing full-length of *fimV, fimV*ᐞᵉ, *fimV*ᐞᶜᶜ, or *fimV*ᐞᶜᴰ contructs. **f** Characteristic morphology of bacteria overexpressing FimVᐞᶜᴰ fused EGFP and the location of FimVᐞᶜᴰ-EGFP in *P. aeruginosa* in vitro and in tissues of wild-type flies. **f′** Quantitative analysis of cell length in (**c**, **f**). **g** FimV-EGFP localization observed by fluorescence microscope in vivo in the hemolymph of wild-type flies. **g′-g″** Quantitative analysis of cell length (**g′**) and width in (**g″**). **h** Survival curves of flies injected with Δ*higA* bacteria overexpressing cell division

genes. **i** Survival curves of flies challenged with PAO1 bacteria overexpressing cell wall elongation genes. **j** Characteristic morphology of PAO1 bacteria overexpressing cell elongation genes in the hemolymph of wild-type or *key* flies. **j′** Quantitative analysis of cell length. Because of the limited number of bacterial cells in most visual fields, all figures in vivo are composites from different visual fields (**b, c, f, g, j**). Bars represent the standard error of the mean (**a, e, h, i**) or the mean (**b′, f′, g′, g″, j′**). Experiments were repeated three times (**a–c, e–j′**; pooled data; **b′**: in vitro ($n = 145$), WT flies ($n = 37$), *key* flies ($n = 37$); **f′**: FimV-EGFP in vitro ($n = 43$), FimV-EGFP in WT flies ($n = 13$), FimVᐞᶜᴰ-EGFP in vitro ($n = 119$), FimVᐞᶜᴰ-EGFP in WT flies ($n = 38$); **g′-g″**: no treatment ($n = 10$), NAG treatment ($n = 10$), NAM treatment ($n = 48$); **j′**: PAO1-*zipA*ᴼᴱ in vitro ($n = 79$), WT flies ($n = 35$), *key* flies ($n = 29$), PAO1-*ftsW*ᴼᴱ in vitro ($n = 26$), WT flies ($n = 36$), *key* flies ($n = 28$)). Statistical analysis was done by Logrank (Mantel-Cox test) in (**a, e, h–j**), by Kruskal-Wallis test with Dunn's post-hoc test in (**b′, f′,g′-g″, i′**). Source data are provided as a Source Data file.

Fig. 9f-f′). Strikingly, the initial polar localization of FimV observed at six hours was lost at 48 hours (Supplementary Fig. 9g-g′), in keeping with the intermediate situation when we inject wild-type PAO1 after overnight culture (60% of bacteria have lost the polar localization of FimV). Furthermore, whereas untreated cells had kept a rod shape, the NAM-exposed bacteria appeared to be roundish (Supplementary Fig. 9g). In vivo, there was no difference between control-injected *fimV*-overexpressing bacteria and those co-injected with NAG at 10 h (Fig. 6g-g″). However, the examination of bacteria retrieved from hemolymph two hours post-injection revealed that FimV enrichment at the bacterial poles was significantly increased in bacteria that had been co-injected with NAG as compared to controls (Supplementary Fig. 9h-h′). Importantly, this finding may account for the faster switch to virulence observed in PAO1 co-injected with NAG. Strikingly, those co-injected with NAM had no enriched localization and were actually roundish with little polarity left (Fig. 6g-g″).

These data demonstrate that NAM interferes with the localization of FimV to the prospective poles and thereby affects the morphology of PAO1.

**The virulence of PAO1 correlates with an elongated morphology**
The results above underscore the importance of bacterial shape and length in virulence. The passage from ellipsoid to bacillus shape involves both cell elongation and cell division since there is a competition for PGN synthesis for lateral cell wall and septum synthesis[49–51]. We therefore tested whether the overexpression of genes required for cell division or PGN synthesis might rescue the Δ*higA* impaired virulence phenotype. The overexpression of genes encoding some components of the divisome (*zipA, ftsQ*, and *ftsW*) or genes required for cell wall elongation (*rodA, mreD*, and *ftsW*) did not alter the Δ*higA* phenotype (Fig. 6h). The morphology of in vitro-cultured Δ*higA* bacteria overexpressing *zipA* was changed from ellipsoid to rod or filaments, suggesting an impact on cell division/septation and lateral wall elongation whereas this change was not observed upon *ftsW* overexpression (Supplementary Fig. 10a). However, both Δ*higA* bacteria overexpressing either *zipA* or *ftsW* acquired a rounded, apolar, rather than an ellipsoid morphology when injected (Supplementary Fig. 10a), in keeping with their low virulence (Table 1). In contrast, the virulence of Δ*higA* bacteria was somewhat increased when some other genes involved in PGN synthesis linked to cell division were overexpressed in this background (*ftsI, ftsL*, and *ftsE/ftsX*) (Supplementary Fig. 10b). However, the virulence of Δ*higA* bacteria was further decreased when the genes encoding divisome components FtsZ (the major structural component of the Z-ring that drives division and a limiting factor for this process), FtsA, or FtsK were overexpressed, almost to avirulence in the case of *ftsZ* overexpression, even though their fitness was not impaired, as documented by their full virulence in *key* mutants (Supplementary Fig. 10c). Thus, this observation establishes that interfering with cell division through *ftsZ* overexpression[52] makes the Δ*higA*

mutants more susceptible to the action of IMD pathway effectors. This correlates again with a rounded morphology of these bacteria in wild-type flies (Supplementary Fig. 10a, Table 1).

All of the genes tested by overexpression in Δ*higA* mutants that either do not alter its virulence or further decrease it displayed a reduced virulence when overexpressed in wild-type PAO1 (Fig. 6i, Supplementary Fig. 10d, e and Table 1). On the one hand, the overexpression of *ftsZ* and *ftsA* in wild-type bacteria led to only a mild reduction in virulence and did not impair their fitness in vivo, as they were able to efficiently kill *key* immunodeficient flies (Supplementary Fig. 10d). Of note, the strong overexpression of *ftsZ* in wild-type *Escherichia coli* bacteria has been reported to block cell division and to lead to filamentation[52], in keeping with our in vitro data (Supplementary Fig. 10a, Table 1). A milder yet similar filamentation phenotype was observed upon *ftsA* overexpression. Interestingly, these bacteria displayed variable morphologies in vivo, from ellipsoid to filamentous (Supplementary Fig. 10a, Table 1). On the other hand, the overexpression of *zipA, rodA, mreD, ftsQ, ftsK*, or *ftsW* led to a reduced virulence in wild-type flies, which was only partially accounted for by their reduced fitness, as monitored in *key* mutants (Fig. 6h, Supplementary Fig. 10e). The reduced virulence was more marked in the case of *zipA* and *mreD* overexpression. In contrast, the overexpression in wild-type PAO1 of the *ftsE, ftsI, ftsL*, or *ftsX* genes, which slightly rescued the virulence defect of Δ*higA* (Supplementary Fig. 10b), hardly altered the virulence of these bacteria (Supplementary Fig. 10f) even though the genes were indeed overexpressed (Supplementary Fig. 10g). Strikingly, bacteria overexpressing *zipA* or *ftsW* formed filaments in vitro, displayed in contrast a rather rounded to ellipsoid morphology when injected into wild-type flies, yet were rod-shaped when injected into *key* flies (Fig. 6j-j′). Thus, the morphology strongly differs between bacteria in vitro and in vivo and an ellipsoid/rounded shape correlates with a moderate or strongly attenuated virulence. Importantly, the overexpression of these divisome components makes them susceptible to the action of the IMD-dependent AMPs that actually prevent the associated elongation observed in vitro or in *key* mutants. Of note, the overexpression of either *ftsZ, rodA, zipA, mreD, ftsA* or *ftsI* does not appear to impair the proliferation/growth of bacteria cultured in vitro (Supplementary Fig. 10h).

We conclude that bacterial shape correlates with virulence in its ability to resist the action of the IMD pathway and that AMPs may interfere with components of the divisome.

## Discussion
In this article, we show that the virulence of planktonic PAO1 bacteria is not constitutive but results from an adaptation phase to the host, markedly by making it able to resist the action of a cocktail of specific antimicrobial peptides. The transition consists in an initial maturation during which the bacteria do not proliferate and switch from an ellipsoid morphology of in vitro cultured bacteria to an elongated bacillus

**Table 1 | Phenotypes of overexpressed cell division and peptidoglycan synthesis genes in wild-type and immuno-deficient flies**

| classification | Gene name | Function | Overexpression in PAO1 | | | | Overexpression in ΔhigA | | | |
|---|---|---|---|---|---|---|---|---|---|---|
| | | | in wt flies | | in key flies | | in wt flies | | in key flies | |
| | | | morphology[a] | virulence | morphology[a] | virulence | morphology[a] | virulence | morphology[a] | virulence |
| septal peptidoglycan synthesis | ftsZ | tubulin structural homologue and master regulator of cell division | ellipsoid/rod | decreased mildly | - | decreased mildly | rounded | decreased | - | unchanged |
| | ftsA | A principal membrane tether for FtsZ | ellipsoid to filamentous | decreased mildly | - | decreased mildly | - | decreased mildly | - | unchanged |
| lateral peptidoglycan synthesis | *mreD* | maintenance rod shape | - | *decreased sharply* | - | *decreased mildly* | - | unchanged | - | unchanged |
| | *zipA* | *PGN polymerase coordination* | *rounded/ellipsoid* | *decreased sharply* | *ellipsoid/ rounded* | *decreased mildly* | rounded | unchanged | - | unchanged |
| | *rodA* | *PGN polymerization GTase* | - | *decreased sharply* | - | *decreased mildly* | - | unchanged | - | unchanged |
| peptidoglycan synthesis | *ftsW* | *PGN polymerization GTase* | *rounded* | *decreased sharply* | *ellipsoid/ rounded* | *decreased mildly* | rounded | unchanged | - | unchanged |
| | *ftsQ* | *recruits FtsWI* | - | *decreased sharply* | - | *decreased mildly* | - | unchanged | - | unchanged |
| | *ftsK* | *interacts with FtsW, FtsQ* | - | *decreased sharply* | - | *decreased mildly* | - | decreased mildly | - | unchanged |
| peptidoglycan synthesis | ftsL | forming trimer with FtsQ and FtsB | - | unchanged | - | unchanged | - | increased mildly | - | unchanged |
| | ftsI | forming dimer with FtsW, transpeptidase | - | unchanged | - | unchanged | - | increased mildly | - | unchanged |
| ATP hydrolysis | ftsE | peptidoglycan autolysin | - | unchanged | - | unchanged | - | increased mildly | - | unchanged |
| | ftsX | peptidoglycan autolysin | - | unchanged | - | unchanged | - | increased mildly | - | unchanged |
| cell wall binding protein | *FimV* | *polar PGN binding-protein* | *rod* | *decreased mildly* | *rod* | *decreased mildly* | *rod* | *increased sharply* | *rod* | *decreased mildly* |

PGN: peptidoglycan, GTase: glycosyl-transferase; -: not determined;
[a]morphology was observed 48 h after injection.

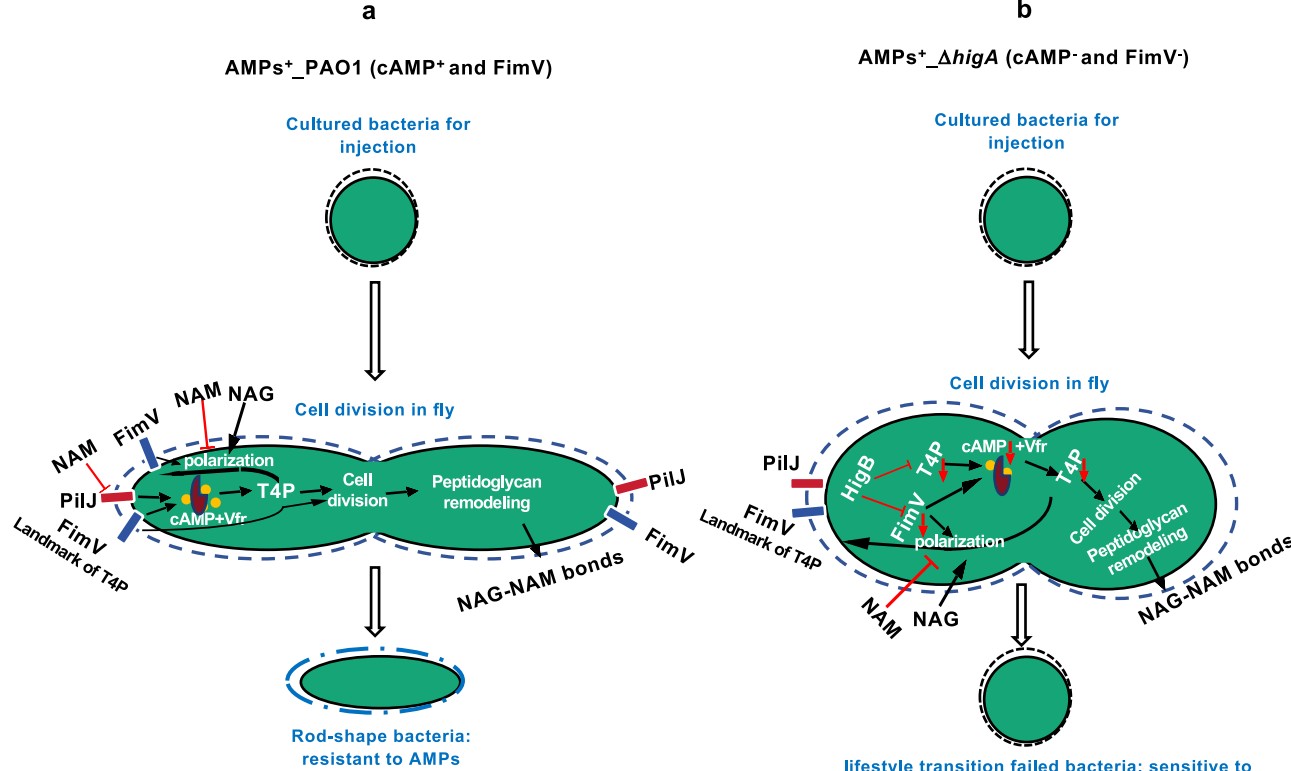

**Fig. 7 | Schematic diagram: N-acetyl-glucosamine primes *P. aeruginosa* by promoting the polarization of *P. aeruginosa* in vivo; the morphology transition protects the bacteria from a set of *Drosophila* antimicrobial peptides.**
**a** N-Acetylglucosamine promotes the polarization of the bacteria by enhancing the localization of FimV, and hence that of the T4P, to the poles of the bacteria. The activation of the cAMP:Vfr axis then stimulates the synthesis of peptidoglycan in the lateral cell wall, thereby ensuring cell elongation. Of note, the expression of the polar protein FimV and type IV pili (T4P) is regulated by the PilJ-cAMP-Vfr signaling cascade in a feed-back loop. N-acetyl-muramic acid prevents the polarization of the bacterial cell by inhibiting the association of FimV to the septal peptidoglycan. Whether the T4P is assembled remains to be determined; yet, Vfr signaling is

prevented at least during early stages of priming. **b** In Δ*higA* bacteria, the toxin HigB reduces the levels of type IV pili (T4P) and the polar protein FimV, which are further repressed due to impaired PilJ-cAMP-Vfr signaling in a positive feedback loop. The Δ*higA* bacteria are only partially polarized and the synthesis of peptidoglycan can be inhibited by antimicrobial peptides that gain access to the peptidoglycan synthesis machinery. AMPs: antimicrobial peptides; cAMP: cyclic adenosine monophosphate, yellow circle dots; PilJ: red bar; FimV: blue bar; Vfr: purple pie-shaped; NAG: N-acetyl-glucosamine; NAM: N-acetylmuramic acid; black solid arrow: promotion; red solid blunt arrow: inhibition; red downward vertical arrow: down-regulated.

morphology. This priming event appears to be promoted by exposure to enzymatically-derived N-acetyl-glucosamine, a likely host-derived signal that also influences morphological switches occurring in fungi[53,54]. Mechanistically, the limited expression of the *higBA* toxin/antitoxin operon is required for the adaptation of the bacteria to its host. The antitoxin HigA prevents the downregulation of the type IV pili apparatus by the HigB toxin. The localization of FimV to the poles of the bacteria is an event of key importance for determining the morphology of the bacteria and hence their virulence; cAMP signaling would then act by promoting the extension of the peptidoglycan lateral cell wall. Thus, an unexpected function of type IV pili is to provide protection against the action of specific AMPs through a FimV/cAMP-mediated modulation of bacterial cell shape (Fig. 7), possibly through sensing indirectly extracellular NAG and not through mechanosensation[37–42].

In contrast to the prevailing view of immediate virulence of pathogenic bacteria in *Drosophila* systemic infection models, *P. aeruginosa* virulence in the host requires an adaptation period, a process that we refer to as priming, during which bacteria do not proliferate for at least 18 h (Fig. 1b). PAO1 bacteria adopt a rod morphology and become resistant to the action of the IMD pathway, which is not yet strongly activated due to a lack of bacterial growth. It is likely that adaptation to the host requires a rewiring of the bacterial metabolism that ultimately enables proliferation in the novel environment. The AMP resistance properties linked to shape changes will be discussed further below.

Interestingly, priming is not the consequence of exposure to the AMP-mediated inducible immune response[55], even though the expression of the *higBA* operon has been reported to be regulated by the CrpRS two-component system upon sensing the mammalian LL-37 AMP[21]. In contrast, here it likely results from glycoside hydrolase enzymatic activities that lead to the release of NAG, such as lysozymes and chitinases. Future studies will determine whether the NAG originates from the host, *e.g.*, degradation of the cuticle exposed during wounding by either host or bacterial chitinases or may be generated by other glycoside hydrolases[56–58]. The possibility of a pathogen bona fide chitinase being involved appears remote since the deletion of its one identified chitinase gene does not alter the virulence properties of *P. aeruginosa*. In contrast, the *Drosophila* genome encodes nine chitinases and no less than 17 lysozymes. Priming may not be limited to *Pseudomonas* and may also exist in the case of some *Providencia* species that also display a delayed virulence[59].

How mechanistically NAG primes PAO1 through PilJ remains to be established since our biochemical and genetic data do not support a direct binding to the ligand binding domains of this chemosensor associated to the T4P. *P. aeruginosa* displays directed twitching towards dioleoyl phosphatidylethanolamine (PE)[60], and it has been reported that the periplasmic domain of PilJ, which contains the two ligand-binding domains, is dispensable for directed twitching motility towards this proposed chemoattractant[61]. It is thus possible that PE

and NAG work through a similar as yet uncharacterized mechanism. One possibility would be an involvement of the transmembrane and HAMP domains as described for the phenol-sensing response by methyl-accepting chemotaxis proteins (MCP) in *Escherichia coli*[62]. Of note, PilJ encodes the sole MCP domain protein known to function in the *P. aeruginosa* chemosensing system.

We have identified two major effectors of the T4P that are both required for bacterial cell shape changes, FimV and Vfr. FimV levels are strongly decreased in Δ*higA* mutants and the overexpression of its gene largely rescued the lessened virulence phenotype of Δ*higA*, provided it is located at the poles through its CC domain. In contrast, *vfr* overexpression failed to rescue the Δ*higA* phenotype, possibly because the remaining low level of FimV in this mutant background is insufficient to compensate the polar information needed for the morphology switch. Our data indicate that NAG does not modulate Vfr signaling. Rather, our data indicate that NAG promotes a faster enrichment of FimV to the poles. NAM prevents the recruitment or stabilization of FimV to the poles, possibly by preventing its binding to PGN as a competitive inhibitor; the finding that NAG is able to counteract the effects of NAM to a sizable extent further confirms that NAG favorizes the localization/maintenance of FimV to the poles.

NAG is an important component of bacterial PGN, fungal chitin, *Myxococcus xanthus* extracellular fibrils, and animal extracellular matrix and cell surface[53,63]. It is becoming increasingly clear that it is involved also in signaling. In this respect, it is striking that NAG promotes the transition of *C. albicans* to a hyphal morphology through cAMP signaling and thereby promotes its virulence[64,65]. Thus, this parallel mechanism likely illustrates yet another example of evolutionary convergence.

Since the co-injection of NAM together with PAO1 essentially abolished its virulence, it will be important to assess whether NAM treatment of burns in human patients may prevent the dramatic colonization of the damaged epithelium by *P. aeruginosa*[66].

One key finding of this report is the link between bacterial shape and length and subsequent susceptibility/resistance to the action of the IMD pathway through three related families of Proline- or Glycine-rich AMPs known to interfere with bacterial translation [Drosocin[67,68]], bacterial membrane permeability and synthesis of outer membrane proteins [Attacins and Diptericins[69–71]]. Interestingly, *Drosophila* Drosocin/Buletin and Diptericins have been shown to be essential for host defense against specific bacterial species[22,72]. How their combination may act in the host defense against *P. aeruginosa* had remained unaddressed until this work. Their combination prevents the extension of the lateral cell wall, for instance that promoted by overexpressing divisome components. Yet, they cannot act upon bacteria that have already an elongated morphology.

Wild-type bacteria grown under our conditions present an initial ellipsoid shape and progressively adopt a rod morphology in vivo independently from the action of the IMD pathway. Importantly, they become progressively longer in vivo, from an initial size of about 1.5 μm in vitro, 2 μm after 13 h of incubation in the fly, to 3 μm at 24 h (Fig. 3a', b', Supplementary Fig. 5a-a'). Vfr is required for the full-size extension as Δ*vfr* bacterial cells were only 2.5 μm-long (Fig. 4g'). We report here a strong correlation between bacterial shape and virulence, with bacteria harboring a coccoid morphology being hardly virulent as is the case for Δ*higA* (see also Table 1). Strikingly, Δ*higA* become much more pathogenic when *fimV* is overexpressed in this background (Fig. 6a), a property that correlates with an elongated rod shape. Δ*higA* mutants are virulent only in immunodeficient flies, where they adapt a rod shape close to 2 μm in length (Fig. 3b'). Interestingly, once primed in *key* flies, but not in wild-type, they are highly virulent when re-injected into wild-type flies and thus, as noted above, able to resist the action of the three sets of AMPs (Fig. 2f, Supplementary Fig. 4g).

Whether our findings are relevant to infection in mammals remains speculative at this stage, albeit our finding that Δ*higA* bacteria

are less virulent in a mouse lung infection model are promising in this regard (Supplementary Fig.11). In addition, the *higBA* operon is found in most clinical isolates[14,73]. It appears that wound infection models in rodents are likely more adapted to studying whether there is any priming occurring also in mammals, especially since multiple AMPs are produced also in their epithelia[74–76]. It remains to be shown whether any of them, or a combination thereof, would target the same process as that we describe here for *Drosophila* Diptericins/Drosocins/Attacins.

## Method

A list of strains and resources is available in Supplementary Data 2.

### Ethical statement

Mice experiments were performed under the guidance of the local South China Agricultural University ethics committee (agreement number 2019C013).

### Drosophila and mouse models

Mated female flies have been used for all *Drosophila* experiments. The wild type fly strain used throughout the experiments is *w[A5001]*, which was generated from re-isogenized *w[1118]* flies[77]. *key, PGRP-LE, PGRP-LC, MyD88* flies are kept in our laboratory[78,79]. *w[A5001]* is the isogenic control flies for the *key, MyD88*, and *PGRP-LC* mutant flies. Some of the fly mutant stocks used in this study have been generated in a *w[1118]* flies background[80]. *PPO1Δ-PPO2Δ* and the series of AMP gene-deficient flies were kind gifts from Bruno Lemaitre[22,81]. p*ubi*-Gal4-p*tub*-Gal80[ts] were constructed from stocks obtained from the Bloomington center stock. Upstream Activating Sequence (UAS)-*imd* flies have been described by Georgel et al.[82] Seven-eight-week-old female SPF C57BL/6 J mice were bought from Jackson Laboratories (Bar Harbor, ME) and raised in the Guangdong Medical Laboratory Animal Center under the following conditions: Specific Pathogen-Free (SPF) conditions; mice were housed at 4~5 per cage with a 12-hour light/dark cycle, and at 25 ± 1°C with 40~70% relative humidity.

### Bacteria strains and culture conditions

The wild type *P. aeruginosa* strain used in this study is the laboratory reference strain PAO1. Mutants used here have been constructed using the suicide vector pEX18AP[18] based on homologous recombination[83]. Details for each mutant are shown in Supplementary Data 2. Plasmids for overexpression of *cyaB, exsA, fimV, ftsZ, ftsA, ftsE, ftsI, ftsK, ftsL, ftsQ, ftsW, ftsX, higB, higA, higBA, mreD, pilJ, rodA, sulA, vfr,* and *zipA* were constructed by cloning the coding sequence from PCR-amplified PAO1 genomic DNA (primers synthesized by Tsingke, China; Supplementary Data 2) into plasmid pMQ70 (ClonExpress MultiS One Step Cloning kit, Vazyme, Nanjing, China) and transformed in the *E. coli* DH5α strain for selection on LB agar plates with 50 μg carbenicillin. Gene overexpression in *P. aeruginosa* was induced by 5 mM L-arabinose in vitro. For inducing bacterial gene overexpression in vivo, flies were fed a solution of 10 mg/mL arabinose in sucrose solution one day prior to the injection of bacteria and kept on this solution for the duration of the experiment. All the bacterial stocks were kept at −80°C; the stocks were retrieved by smearing on LB agar plate and cultured at 37°C overnight. Single fresh colonies were picked to inoculate the BHI broth overnight and exponential phase bacterial cultures were harvested by centrifugation. The bacterial pellet was then resuspended and washed in phosphate-buffered saline twice.

### Acute infection

The harvested *P. aeruginosa* pellets were suspended in PBS to measure their optical densities and adjusted to OD600 1.0 in PBS and then diluted 100-fold for injection. 3-7-day-old female adult flies were picked at random for injection, and then 13.8nL of the bacterial suspension (about 100 bacteria, unless otherwise indicated) was injected into the thorax of each fly using the Nanoject III (Drummond, 3-000-

032, USA). The infected flies were incubated at 18°C with 60% humidity in a climate chamber and counted every 12 hours until flies died out.

## Latent infection

Before infection, 3-7-day-old female adult flies were picked for pre-treatment by feeding them on a 100mM sucrose solution deposited on a Millipore pad for 2 days at 25 °C. The harvested *P. aeruginosa* pellets were suspended into 100 mM sucrose solution and the optical density of the bacterial suspension was adjusted to $OD_{600}$ 10.0 in sucrose solution for the next infection experiment. Each tube with 20 flies was exposed to 600 μL bacterial solution on a Millipore filter and placed at 18 °C for 2 days. Then, the infected flies by ingestion were transferred to a new tube with 600 μL of 100 mM sucrose solution supplemented with 100 μg gentamicin on a Millipore filter and kept for 4 days at 18 °C in this vial to kill *P. aeruginosa* cells in the gut lumen. Next, the flies were transferred to new tubes with a filter soaked with 600 μL of 100 mM sucrose solution only. The flies were counted at regular intervals until flies died out.

## Acute lung infection in mice

*P. aeruginosa* strains were grown on BHB for with shaking (220 r.p.m.) at 37 °C for 16 - 18 h, then washed and suspended in PBS, and diluted to OD600 0.1 in a volume of 50 μL. C57BL/6 J (7- to 8-week-old) females were anesthetized by diethyl ether and were infected by non-invasive intratracheal instillation of PAO1 at the indicated dose. Groups of about 10 mice were monitored over 7 days for survival whereas bacterial load was measured as described further below in blood, lung, liver, and kidneys 36 h post-infection. Animals were humanely euthanized when they appeared to be under acute distress and counted as diseased.

## Bacterial titer in hemolymph

The empty capillary needle was fixed into the Nanoject III machine (Drummond, 3-000-032, USA), then was pricked on the thorax of each fly to collect hemolymph on the basis of a capillarity extraction effect. The collected hemolymph from each fly was diluted into 10 μL pre-pared PBS in 1.5 mL Eppendorf tube. Samples collected above were diluted by a series of 10-fold dilution and then plated on LB agar. These plates were put in an incubator at 37 °C overnight to count the colony-forming units.

## Bacterial titer in whole flies

Each anesthetized fly was put into 1.5 ml Eppendorf tube with 50ul PBS inside and then crushed the flies using the Mixer Mill MM400 (Retsch, Germany) at 30Hertz for 10 sec. The samples prepared above were diluted by a series of 10-fold dilution and then plated on LB agar and incubated at 37 °C overnight to count the colony forming units.

## Bacterial titer in mice organs

Blood from anesthetized infected mice was collected by the orbital sinus method prior to euthanasia at 36 h post-infection; their organs were then collected aseptically, weighed, and homogenized in 1 mL of PBS. Tissue homogenates were serially diluted and plated on 100 μg/mL ampicillin LB-agar plates and colony forming units (CFU) counting was made 16–18 h later.

## RNA extraction and reverse transcription for RT-qPCR

RNA extraction and reverse transcription was performed as described earlier[9]. Gene transcription level in flies and bacterial cells was measured by reverse quantitative PCR using a SYBR Green fluorescent dye. Primers used for quantification were synthesized by Tsingke (China) are shown in Supplementary Data 3. The 5x-diluted RT samples for detection were mixed with ChamQ SYBR qPCR Master Mix according to supplier instructions and then run on CFX 96 or CFX 384 (Biorad, USA). The expression level of target genes was normalized by the $2^{-\Delta\Delta Ct}$ method against *Rpl32* (encoding ribosomal protein 49) transcripts

(flies) or *oprL*/16 s (*P. aeruginosa*) after having checked that the efficiencies of the primer couples were similar. For Supplementary Fig. 3, the AMP genes steady-state transcription levels normalized to bacterial load were measured using the carcass of single flies from which hemolymph had been collected to measure the bacterial load. To measure bacterial gene transcription at early infection stages, up to 10 h, a higher bacterial dose (10^5 CFU/fly) had been inoculated into flies (Fig. 5f, 5i, 5k). In other cases, bacterial gene transcription was measured under standardized infection conditions.

## Co-injection of PAO1 and ΔhigA cells

The overnight cultured PAO1 and ΔhigA bacterial cells were harvested, washed and diluted as for acute infection. These two diluted bacterial cells were mixed 1:1 for injection to make sure that this mixture has the same overall number of bacterial cells as the PAO1 or ΔhigA control groups.

## Acute infection in *imd*-overexpression flies

Virgin UAS-*imd* flies were crossed to p*ubi*-Gal4-p*tub*-Gal80$^{ts}$ males. The crossed flies were put at 18 °C and the offspring developed at 18 °C into adult flies. The *imd* gene was overexpressed by placing flies at 29 °C for at least 5 days before injection procedure. The survival analysis of injected flies by *P. aeruginosa* was performed at 29 °C.

## Heat killed *P. aeruginosa* injection

The cultured *P. aeruginosa* were harvested and suspended in PBS to be adjusted into $OD_{600}$ 10 and boiled at 95 °C for 30 min. The heat-killed bacteria solution was plated on LB agar to verify there no live bacteria remained. 13.8nL of heat-killed bacteria solution was injected into adult female flies.

## Secondary acute infection

The cultured GFP-expressing *P. aeruginosa* were harvested and suspended in PBS to an OD600 of 0.01, and 13.8 nL of the suspension was injected into flies as a first challenge. The flies were put in a climate chamber at 18 °C for 24 hours, then an RFP-expressing *P. aeruginosa* suspension of $OD_{600}$ 0.1 was injected into the GFP-bacteria injection recipient flies. The death of flies was subsequently recorded every 12 hours or flies were sacrificed to measure their bacterial burden at different time points before death. The bacterial colonies were observed under a fluorescence microscope to discriminate between GFP- and RFP-expressing bacteria.

## Bacteria retrieved from hemolymph for injection

The empty capillary needle was fixed into the Nanoject III machine (Drummond, 3-000-032, USA), then was pricked on the thorax of each fly to collect hemolymph on the basis of a capillarity extraction effect. The retrieved hemolymph of a single fly was immediately reinjected into a single naive fly or a fly pre-injected with heat-killed bacteria 24 h earlier.

## Cell wall disruption bacterial cells injection

The overnight-grown bacterial cells were harvested by centrifugation as described above and washed with PBS twice. The bacterial pellet was suspended in digestion buffer containing 100 μM EDTA (or not) and 100 ng/mL lysozyme for 30 min, then pelleted by centrifugation at 2680 g. The bacterial cell pellet was washed in PBS twice and then its concentration was adjusted in PBS as required prior to injection.

## NAG/NAM injection assay

For NAG/NAM co-injection experiments, the overnight grown bacterial cells were harvested, washed, and adjusted at $OD_{600}$ 0.01PBS containing NAG-NAM, NAG, or NAM solution (10 mg/mL) for immediate injection. Actually, similar results were obtained when the NAG or NAM was pre-injected into flies 30 min prior to the injection of the bacterial

solution. For assessing the role of NAG/NAM on primed bacteria, NAG/NAM was injected into flies that had been injected with bacteria 24 h earlier. To examine the effects of NAG and NAM on FimV-EGFP localization in bacteria in vivo, flies were inoculated with a higher bacterial dose (10^5 CFU/fly) mixed with either NAG or NAM (Fig. 6g).

## Chitinase injection experiment

The bacteria-produced chitinase powder (Macklin, Shanghai, China) was dissolved in water at 25 units/mL and 13.8 nL were then injected into flies. These flies were subsequently injected with *P. aeruginosa* suspension about one hour thereafter.

## Lysozyme digestion assay

The overnight cultured bacteria were harvested and resuspended in PBS, and were then mixed if needed with different doses of NAG/NAM or their mixture in a 500 µg/mL lysozyme solution in 96-well plate. The plate was put at 37 °C. The turbidity of bacteria was measured by the microplate reader spectrophotometer at 600 nm.

## Transmission electron microscope assay and morphology assay by O-antigen staining

Bacteria observation by transmission electron microscope was performed as described[9]. Briefly, we collected hemolymph of flies containing bacteria; the samples were then observed and images taken with a transmission electron microscope, without fixation, embedding or sectioning. The morphology of bacteria was also observed in some experiments by fluorescence microscopy after O-antigen staining. In brief, the hemolymph with circulating bacteria was collected using the Nanoject III with empty needles based on capillarity extraction. The bacteria associated with tissues were collected just by dissecting infected flies to obtain tissues. The fly tissue with adhering bacteria were fixed with 4% paraformaldehyde for 30 min and then washed with PBS solution three times. The samples were firstly incubated with 5% BSA solution for 2 h, then incubated with the O5 primary antibody (Biorbyt, orb234239) diluted 1:1000 in 5% BSA solution for 1 h. The samples were washed in PBS for 3 times, and then incubated with the DyLight™−488 labeled fluorescent secondary antibody (Thermo Scientific, 35502) diluted 1:5000 raised against mice Ig for 1 h. The samples were washed in PBS three times and then observed under fluorescence microscopy. The length of bacterial cells was quantified using Microbe J. Of note, in some cases to improve the quality of pictures, bacteria were not directly photographed in the tissues but on collected hemolymph, as indicated. We have, however, first checked that the shape of bacteria was qualitatively and quantitatively similar in tissues and in collected hemolymph, albeit with lower sampling in the tissues. Fluorescence images were acquired on a ZEISS Axio Imager M2 system (100× oil objective) using 488 nm excitation and 525 nm emission to observe the 488-conjugated antibody staining bacteria and fluorescence green protein labeled bacteria. Raw files were processed in ZEN. Lnk and Fiji.

## Proteomic analysis

Wild type PAO1, Δ*higA*, Δ*higB*, Δ*higBA* mutants (each in three biological triplicates) cultured in BHI were harvested by centrifugation at exponential phase and put in dry ice for delivery to the Applied Protein Technology company (Shanghai, China). A total of twelve samples were lysed using SDT buffer (4% SDS, 100 mM Tris-HCl, 1 mM DTT, pH 7.6). Protein amount was determined by BCA assay kit (Beyotime, P0012). Proteins (200 µg/sample) were dissolved in 30 µl SDT buffer (4% SDS, 100 mM DTT, 150 mM Tris-HCl pH 8.0), detergent and small molecules in which were removed by ultrafiltration (Microcon units, 10 kDa cutoff). Cysteines were alkylated with 100 mM iodoacetamide following by UA buffer washes (3×) and NH$_4$HCO$_3$ washes (2×). Trypsin (Beijing Hualishi Biotech, HLS TRY001C) digestion (4 µg, 37 °C/overnight) was performed in 25 mM NH$_4$HCO$_3$. Peptides were desalted (Empore™ SPE Cartridges C18, bed I.D. 7 mm, volume 3 ml, Thermo Fisher), vacuum-concentrated, and resuspended in 0.1% formic acid, which was determined by A$_{280}$. Peptides (100 µg/sample) were labeled with TMT reagents (Thermo Scientific) following manufacturer protocols. Labeled peptides were fractionated into 10 fractions using a High pH Reversed-Phase Peptide Fractionation Kit (Thermo Scientific), then desalted and concentrated. Peptides were loaded onto a reverse phase trap column (Thermo Scientific Acclaim PepMap100, 100 µm*2 cm, nanoViper C18) connected to the C18-reversed phase analytical column (Thermo Scientific Easy Column, 10 cm long, 75 µm inner diameter, 3µm resin) in buffer A (0.1% Formic acid) and separated with a linear gradient of buffer B (84% acetonitrile and 0.1% Formic acid) at a flow rate of 300 nl/min controlled by IntelliFlow technology. MS data were acquired on a Q Exactive mass spectrometer (Thermo Scientific) in data-dependent mode: Full scans (300–1800 m/z) at 70,000 resolution (m/z 200), followed by HCD fragmentation of the top 10 precursors (17,500 resolution, 2 m/z isolation width, 30 eV collision energy). Automatic gain control was set to 3e6 with 10 ms maximum injection time. Raw files were analyzed using the MASCOT engine (Matrix Science, London, UK, v2.2) embedded into Proteome Discoverer 1.4 software for identification and quantitation analysis. Search parameters: Trypsin (max missed cleavages: 2); fixed modifications (TMT (Tandem Mass Tag) 16-plex (N-terminus/K)); variable modifications (TMT 16-plex (Y)); mass tolerances(±20ppm); fragment mass tolerant (0.1 Da); database pattern (uniprot_Pseudomonas_aeruginosa_50503_20210715.fasta). The decoy database approach controlled for FDR (≤1% at peptide level). Protein quantification used median ratios of unique peptides, and the minimum peptide length for detection is 6. Proteins with fold change above or below 1.2 and *p*-value less than 0.05 were identified as being differentially expressed among mutants and wild type PAO1 group. Bioinformatics analysis included GO/KEGG annotation (Blast2GO), PPI networks (STRING/Cytoscape v3.2.1), and hierarchical clustering (Cluster 3.0).

## Bacterial growth assay in vitro

To measure the growth of the set of *highBA* mutants as compared to their proliferation in vivo, the overnight cultures of these bacterial strains were inoculated into 30mL BHB solution in a conical flask and then placed at 18 °C. Samples were taken from the bacterial solution at different time points and diluted to a plate for colony counting. To investigate the role of NAG and NAM in bacterial growth in vitro, the overnight culture of PAO1 solution was diluted 1000-fold and incubated with different doses of NAG/NAM and cultured in 96-well plates at 37 °C for 12 hours. The turbidity was measured using a microplate reader spectrophotometer at 600 nm. To determine the role of cell division genes overexpression on bacterial growth in vitro, the overnight culture of some bacterial strains was inoculated in LB broth (1:100) supplemented with 50 µg/mL carbenicillin in 12-well plates for overnight culture at 37 °C. The turbidity was measured using a microplate reader spectrophotometer at 600 nm. To investigate the effects of NAG and NAM on cell elongation in the FimV-EGFP-overexpressing bacterial strain, 5 mg/mL of each compound was supplemented into the culture medium, and bacterial morphology was examined at different time points post-inoculation of the cultures.

## Twitching assay

This experiment was performed according to the published protocol[84]. In brief, the overnight bacterial culture was adjusted to 1 OD$_{600}$ in LB broth and 10 µL bacterial solution was deposited on the surface of 1% agar plates, which were incubated overnight at 37 °C. The interstitial colony was visualized by flooding the plates with the twitching motility developer solution containing 10% glacial acetic acid and 50% methanol.

## Protein expression and purification

The open reading frame of the target gene PilJ and its ligand binding domain (LBD, residues 39-303)[45] were amplified and cloned into the

protein expression vector pET28a for sequencing. E. coli BL21 cells harbouring recombinant plasmids were cultured in LB broth and induced by IPTG at the optimal concentration at 30 °C. The bacterial cells were then harvested by centrifugation and resuspended in binding buffer with 100 μg/ml lysozyme prior to lysis performed using an ultrasonic cell crusher. The recombinant proteins with His-tag were purified according to instruction of ProteinIso® Ni-NTA Resin. The concentration and purification of these proteins were checked by SDS-PAGE and Coomassie brilliant blue staining.

## Thermal shift assay
The salt ions were removed from the purified protein solution using Centricon filters. The proteins were incubated with potential ligands NAG (90 μM) or NAM (30 μM) in the solution buffer with diluted SYPRO Orange and then the plate was read using a Bio-Rad RT-qPCR machine as previously described according to instructions[85].

## Surface contact activation assay
The overnight cultured PAO1 in LB was harvested and resuspended in PBS before being deposited on the surface of 1.5% agarose plate and incubated for 5 hours. The bacteria were then harvested and checked either for the induction of the *xphA* gene using RTqPCR and normalized to16S RNA or to perform infection experiment using planktonic bacterial cells as controls.

## Reagents and software
All the reagents and software used in this study are listed in Supplementary Data 4.

## Statistical analysis
All the experiments in this study were performed at least three times unless mentioned otherwise and the graphs in the figure panels display the pooled data, which were analyzed using the Prism Graphpad 6.0 software. The survival curves were analyzed by the Logrank (Mantel-Cox test) and the bacterial load sets were analyzed according to their distribution. Kruskal-Wallis test with Dunn's post-hoc test were used for multiple comparisons as indicated and the Student's t-test was used for comparison between two groups. *, $p < 0.05$; **, $p < 0.01$; ***, $p < 0.001$; ****, $p < 0.0001$. n.s., not significant.

## Reporting summary
Further information on research design is available in the Nature Portfolio Reporting Summary linked to this article.

# Data availability
The proteomic data generated in this study have been deposited in PRIDE (PRoteomics IDEntifications Database) under accession code (PXD060361). The data are publicly available without restrictions. Source data have been deposited in figshare at https://doi.org/10.6084/m9.figshare.25913203.v1 as Prism files and are also available as a Source data Excel file.

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

## Acknowledgements

We are grateful to Dr. Bruno Lemaitre for the kind gifts of $PPO1^{\Delta}$-$PPO2^{\Delta}$ and AMP genes deficient fly strains. We acknowledge the technical help of Xiaomin Chang. We thank Drs. Frederick Ausubel and Ivo Gomperts-Boneca for constructive comments on an earlier version of this manuscript. This work was supported by the China Postdoctoral Science Foundation (2017M612634), Special Fund for Scientific and Technological Innovation Strategy of Guangdong Province (2018A030310180), and Scientific Research Capacity Improvement Project of Guangzhou Medical University (02-410-2302286XM) to J.C., by grants from the 111 Project (#D18010; China), the Incubation Project for Innovative Teams of the Guangzhou Medical University, the Open Project from State Key Laboratory of Respiratory Diseases, China, the China High-end Foreign Talent Program, and NSFC (#32370931) to D.F.

## Author contributions

Conceptualization: J.C., G.L., X.W., and D.F.; Methodology: J.C. and G.L.; Investigation and Formal Analysis: J.C., G.L., and K.M.; Resources: J.C., G.L., K.M., Y.G., and X.W.; Writing-Original draft: J.C., G.L., and D.F.; Writing- Review and Editing: J.C., G.L., Y.G., X.W., and D.F.; Visualization: J.C., G.L., and K.M.; Supervision: Z.L. and D.F.; Project Administration and Funding Acquisition: J.C., Z.L., and D.F.

## Competing interests

The authors declare no competing interests.
