## [Transparent Peer Review file · Nature Communications]

N-acetyl-glucosamine primes *Pseudomonas aeruginosa* for virulence through a type IV pili/cAMP-mediated morphology transition

Corresponding Author: Professor Dominique Ferrandon

Version 0:

Reviewer comments:

Reviewer #1

(Remarks to the Author)

In this manuscript, the authors proposed a new concept that “virulence is acquired through a priming process.” using a *Pseudomonas aeruginosa* infection model. They claimed that priming can be initiated by exposure to extracellular N-acetyl-glucosamine (NAG), which is sensed through the type IV pili (TFP) chemosensor PilJ that activates cAMP signaling. While this story sounds interesting, the conclusion is not very solid based their current experiment settings. I have the following major points:

I)The authors claimed that priming first allows the bacteria to proliferate actively and second allows them to withstand the action of the IMD pathway, likely by resisting the combined antimicrobial action of the two/three AMP families. HigB toxin makes the bacteria vulnerable to a combination of AMPs belonging to two/three families. It is well known that *anr* and *pmr* operons are induced by AMP treatment in *P. aeruginosa*, do these operons get induced in vivo? If so, when these operons get induced after infection?

II)The pre-injection of a chitinase into flies minutes prior to a challenge with PAO1 similarly led to enhanced virulence of the injected bacteria. The authors concluded that NAG likely originating from the host promotes the transition to a bacillus morphology, which correlates with virulence. *P. aeruginosa* quorum sensing is known to regulated chitinase production (PMID:11717261). According to this reference “after overnight growth, the majority of the ChiC produced was found intracellularly, whereas only small amounts were detected in the culture medium. However, after several days, the cellular pool of ChiC was largely depleted, and the protein was found in the culture medium.” Thus it is likely a chitinase-like protease is released from *P. aeruginosa* slowly and which is able to generate NAG from host. Do QS get induced during this infection model?

III)To better understand the role of the *higBA* operon, they authors assessed the quantitative expression of PAO1 proteins in Δ *higA*, Δ *higB*, and Δ *higBA* bacteria relative to those found in wild-type bacteria. However, the authors only did in vitro proteomics analysis as documented in their method part “Wild type PAO1, Δ *higA*, Δ *higB*, Δ *higBA* mutants (each in biological triplicates) cultured in BHI were harvested by centrifugation at exponential phase and put in dry ice for delivery to the Applied Protein Technology company (Shanghai, China).” Since the authors are able to isolate primed bacteria in vivo after infection, why not comparing the proteome of in vivo isolated primed bacteria with naïve bacteria of the above strains. We all know in vitro and in vivo *P. aeruginosa* have distinct transcriptome and proteome (e.g. PMID: 35913171).

VI) The authors suggested that PilJ may be sensing NAG but not NAM. This hypothesis is very weak. Pilus is important for biofilm formation and colonization in vivo of *P. aeruginosa*. Type VI pili is also required for full level of QS in biofilm mode of growth. A in-depth biochemical characterization is required to test whether PilJ is able to interact with NAG such as ITC experiment.

V) The authors next examined many genes in the cAMP signaling pathway and the divisome and elongation. This part is not closed linked to the *higBA* operon since their proteomics study does not show significant expression changes of these genes. In fact, the authors written in the introduction “HigA in addition is also able to bind to DNA and functions as a repressor, not only of *higBA* itself but also in the promoters of genes involved in virulence such as *mvfR* or type 3 and 6

secretion systems." And T3SS and mvfR are both published important for infections in different animal models. Thus, again the authors should perform the proteome of in vivo isolated primed bacteria with naïve bacteria.

Reviewer #2

(Remarks to the Author)

I think there are some interesting results in this study, but more experiments and analyses are needed to characterize the morphological change that occurs during infection. There are also several misconceptions about virulence and biofilm formation in *Pseudomonas aeruginosa* that are stated several times throughout the manuscript. Many citations are missing that could better frame results and improve the design of several experiments. Much of the observed data could likely be described via surface-dependent cAMP activation and cellular memory.

-lines 49-53: Both acute virulence and biofilm formation require surface contact and surface sensing via Type IV pili in *Pseudomonas aeruginosa*. Surface sensing triggers cAMP activation via type IV pili interaction with the surface leading to activation of the global virulence regulator, Vfr, which is critical for expression of T2SS and T3SS PMID: 26041805, 37382531, 20494996, and many others. Type IV pili interaction with the surface also triggers c-di-GMP production leading to biofilm formation PMID: 25626906, 34168081. Furthermore, type IV pili are critical virulence factors and allow for motility within epithelial cells likely triggering cAMP production PMID: 31431558. cAMP activation due to type iv pili surface sensing has also been shown to persist for multiple generations within *Pseudomonas aeruginosa* and serve as a molecular memory of surface contact which explains the continued virulence of cells after being recovered from the first infection PMID: 29559526. Also, the extent of cAMP signaling relates to the stiffness of the material to which type IV pili are attached to PMID: 35561220, which is a maximum when binding to surfaces with a stiffness similar to that of human tissues.

-For all figures reporting CFU data, please change the y-axis to log₁₀ CFU/ml.

-Fig1. Instead of adding 1 to your Cfu data to avoid you can state "below limit of detection" or "0 CFU".

-lines 106-107: the important control here is that higA and the higB OE strain grow at the same rate as WT. Growth curves are typically done via 24-hour growth curves with many time points characterizing each phase of the growth curve, not 5 time points over five days. There is no description of the in vitro growth conditions in the methods.

-lines 136-138 reference "a lack of bacterial growth until at least 18h (Fig. S3E)," but fig S3E is a survival curve of flies and does not show an increase in bacterial CFU over time which would be indicative of growth.

-lines 159-166: PMID: 29559526

-line 167: please explain how "priming" is different or distinct from any other typical lag phases in a new growth medium.

-line 175-180/Fig3A-C: There needs to be some sort of quantification with some sort of image analysis software to conclude that there is any significant difference in cell shape/morphology. Without any sort of quantification and statistical analysis I am not comfortable making any sort of conclusion. Reference 9 also lacks any sort of statistical image analysis of the morphological changes of bacteria. Furthermore, if the shape of bacterial cells are to be evaluated, they should all be in the same focal plane.

-all overexpression experiments should include westerns to show stable overexpression of the protein.

-Figure S5E: there seems to be a significant difference between Nam_higA strain w/ 0mg/ml and 10mg/ml but there are not statistical tests to tell if anything is different.

-Fig 4C: please confirm the overexpression of PilJ with a western.

-Line 271: One of the few ligands that actually bind to the ligand binding domain of PilJ are phenol soluble modulins produced by *Staph aureus*. To properly test if PilJ is sensing NAG, you should use PilJ with the ligand binding domain removed which has been used in several studies so far: PMID 27190147, 38349934. Without this control you cannot conclude that PilJ is sensing NAG and not just transducing a surface signal from type IV pili.

-line 277: the citations listed do not connect cAMP levels with cell wall remodeling

-lines 273-287: Even though overexpression of vfr does not compensate virulence it does not mean Vfr does not have a downstream role in virulence given its function. Vfr has a cAMP dependent and cAMP independent regulon PMID: 20494996. Overexpression of Vfr without an equal activation of cAMP means it is only activating the non-cAMP regulon which are not virulence genes. Furthermore, Vfr is the only known cAMP binding protein in *Pseudomonas* with any physiological effect. PMID 38832786. The other mutants indicate Vfr likely plays a role in a cAMP-dependent manner.

-Fig 5E: There should be quantification of polar localization like using microbeJ or Fiji. FimV appears to be cytoplasmic when OE in liquid conditions. Surface sensing leads to an increase in the number of type iv pili as well as localization of pili

components.

-line 313-316: The data do not support the claim made. These mutations are supposed to make the cell morphology more rod like and therefore virulent but in line 308-312 you state that overexpression leads to decreased virulence.

-Type IV pili interact with surfaces and are able to bind surfaces better with EPS. Perhaps NAG can bind type iv pili and mimic a surface that would trigger Pil-Chp, leading to cAMP production, Vfr-cAMP dependent regulon, and increased virulence. This positive feedback loop maintains high levels of cAMP over multiple generations and is a hypothesis worth testing here.

-line 352: mechanosensing is not tested in this manuscript so this claim cannot be made.

-line 355-357: many papers have already shown this to not be true.

-line 383-387: The claim about bacterial shape and resistance to the IMD pathway is not supported by the last set of experiments in which mutants exhibiting more rod-like morphology are still cleared by IMD.

-FimV is necessary for cAMP activation via type iv pili and flagellar mediated surface sensing so it makes sense increased FimV increases virulence phenotypes. PMID: 31682637

Reviewer #3

(Remarks to the Author)

The present work describes the process by which the bacterial pathogen *Pseudomonas aeruginosa* adapts to the host environment in the initial stage of the infection. Using *Drosophila melanogaster* as their acute infection model, this transition period allows the bacteria to not proliferate, hiding from the host's immune system, and switching from an ellipsoid shape to a rod morphology. The work clearly shows a correlation between the bacillus morphology and virulence. Furthermore, the authors also elucidate the probable mechanism, from the extracellular signal required to initiate the transition to the genes related to bacteria wall remodeling. Overall, the data is solid with some minor concerns over significance in terms of extrapolating to mammalian infection models. The authors should address this in more detail in the discussion especially in light of the use of antimicrobial peptides.

Despite this the work clearly helps expand the understanding about the molecular mechanisms involved in pathogen adaptation to the host during different stages of infection, here specifically during the acute stage. This knowledge is crucial to understand the process of colonization that results from bacterial infections and for the development of new therapies and treatments targeting bacterial pathogens characterized as multi-drug resistant, such as *Pseudomonas aeruginosa*.

The design and analysis of a wide variety of mutants in *P. aeruginosa* and on the side of the host (flies) is an innovative allowing a mechanistic teasing out of the role of the NAG in remodelling cell morphology and priming of virulence. They clearly show the sensing of NAG is a common signal that promotes virulence through morphology changes. However, the authors might consider higher level imaging such as electron microscopy to confirm the morphology experiments as it is difficult to really judge the morphologies. Also, in Fig 3A-C more detail of the conditions and time frame for the images would be helpful to the reader. The priming phase by NAG is mediated by the T4P chemosensing system by activating downstream signaling via cAMP mediated cell wall remodeling. The authors make the case that morphology drives virulence. Overall the technical quality of the manuscript is extremely high and the results consistent with the conclusions. Although the experiments as performed are of a high quality the text needs to be improved in consistency and clarity. Specifically, I encourage the authors to review the consistency between the names listed in the tables and the text and figures, as some of them do not match or are not shown in the tables. Then, I would recommend including a complete list of the plasmids designed and/or assayed as well as for the bacteria or fly mutants. In addition to just the reference, including a brief description of the constructs, such as the plasmid backbone, inducible or constitutive promoter, gene/pathway affected, etc, would be recommended and would improve the understanding of the text.

Overall, this is a rigorous and interesting report on *P. aeruginosa* priming and adaptation in the *D. melanogaster* model system. The results are significant in providing a foundation for the role of priming in an acute infection and open further avenues of research in other model systems.

Reviewer #4

(Remarks to the Author)

Version 1:

Reviewer comments:

Reviewer #1

(Remarks to the Author)

The study investigates the role of N-acetyl-glucosamine (NAG) in priming *Pseudomonas aeruginosa* for virulence via a T4P/cAMP-mediated morphological transition. While the revised manuscript addresses some concerns and presents novel insights into host-pathogen interactions, several critical issues remain unresolved. Below are detailed critiques to strengthen the scientific rigor and clarity of the work:

Major Concerns

1. Image Quality and Presentation

- Several fluorescence microscopy images lack clarity (e.g., Fig. 3C, 3D, 4G, 5B–C). The use of composite panels from disparate fields creates confusion, and scale bars are inconsistently placed or absent. The authors should provide single-field, high-resolution images with uniform scale bars (e.g., 2 μ m) clearly labeled in all panels. Also, please include detailed figure legends explaining image stitching (if necessary) and statistical significance (e.g., n-values, replicates). Standardize annotations are also required (e.g., replace “wt” with “WT,” remove underscores in labels like “WT_?”).

2. Mechanistic Evidence for NAG Sensing by PilJ

- The claim that PilJ indirectly senses NAG is weakly supported. While the thermal shift assay (Fig. 4L) shows no direct binding of NAG to the PilJ LBD, no alternative mechanism is proposed. The authors should measure cAMP levels in Δ pilJ mutants exposed to NAG to test whether NAG activates cAMP signaling in a PilJ-dependent manner. They should perform genetic complementation with PilJ truncation mutants (e.g., Δ LBD) to confirm if NAG sensing requires other domains (e.g., transmembrane or HAMP domains).

3. Lack of Validation in Mammalian Models

The study relies solely on *Drosophila* infection assays. The relevance of the proposed NAG/PilJ/cAMP/FimV axis in mammalian models (e.g., murine acute lung infection) remains unaddressed. They could test virulence of Δ higA*, Δ pilJ*, or NAG-treated PAO1 in a mouse model (e.g., intratracheal infection). If animal experiments are infeasible, explicitly discuss the limitations of extrapolating findings from flies to mammals.

4. Absence of Protein-Level Validation

The key molecular mechanisms (e.g., FimV localization, PilJ expression in truncation mutants) lack validation via Western blotting or immunostaining. The authors should confirm FimV protein expression and polar localization using anti-FimV antibodies (commercially available or custom-designed). They should validate expression levels of truncated PilJ constructs (Fig. 4M) via Western blot to ensure genetic complementation is functional.

5. Quantification of Morphological Data

The central claim linking bacillus morphology to virulence relies on qualitative images (e.g., Fig. 3A–C, 4G–K) without robust quantification. The authors could use automated tools (e.g., CellProfiler, MicrobeJ) to quantify bacterial length, width, and aspect ratio across ≥ 100 cells per condition, and include statistical comparisons (e.g., ANOVA with post-hoc tests) between groups (e.g., WT vs. Δ higA in key flies).

6. Fitness of Overexpression Strains

The partial rescue of Δ higA virulence by pilJ, cyaB, or fimV overexpression (Figs. 4C–D, 5A) may reflect fitness trade-offs rather than mechanistic effects. The authors should perform growth curves or competitive index assays in vitro and in vivo to rule out fitness costs.

7. Unresolved Mechanism of NAM Toxicity

The observation that NAM abolishes virulence (Fig. 3I) is intriguing but unexplained.

Minor Concerns

1. Figure Standardization

- Color schemes: Adopt consistent color coding across all figures (e.g., use Fig. 1B’s orange/green/blue/red scheme for WT/mutant/rescue groups).

- Axis labels:

- Remove redundant labels (e.g., “10” in Fig. 1A x-axis).

- Use uniform terminology (e.g., “Bacterial load (CFU/fly)” in Fig. 1B, 2C).

- Fonts and symbols:

- Standardize font sizes and styles (e.g., Fig. 3B’ uses “PAO1” in a different font).

- Replace “ns” with “n.s.” (not significant) and ensure asterisks align with statistical thresholds.

2. Scheme Clarity

- Simplify and standardize schematics (e.g., Fig. 3J, 4H, 5G). Move overly simplistic diagrams to supplementary materials.

3. Technical Details

- Clarify ambiguous labels (e.g., “rpl49” in Fig. 2A should be defined as a normalization control).

- Correct distorted fonts (e.g., Fig. 4L’) and align panel labels (e.g., “WT” vs. “key” in Fig. 3B’).

Reviewer #2

(Remarks to the Author)

The authors have performed a substantial amount of work and quantification to improve the manuscript and have adequately addressed each of my concerns. Below are minor comments:

- Recent studies have shown a significant difference in surface sensing, colonization, and biofilm formation between PAO1 and PA14, specifically with Wsp/c-di-GMP being much more important for PAO1 and Pil-Chp/cAMP being more important for PA14 PMID: 34516283, 32098815. This is one of the first studies to highlight Pil-Chp in PAO1, and I wonder how these experiments would differ if PA14 were used, although this is out of the scope of this study.

- line 321-324 change PilJ gene to lowercase

Reviewer #3

(Remarks to the Author)

The authors have performed a significant number of extra experiments to address the major concerns including extra qPCR experiments to ensure and per are not induced on AMP treatment, generation of a chic mutant that has a similar phenotype to WT indicating the NAG is not generated by the host proteases. Moreover, they tested the cAMP activation and memory with additional experiments to show priming is affected independent of surface sensing. Overall the manuscript is vastly improved and contributes to our understanding of virulence priming by NAG.

Reviewer #4

(Remarks to the Author)

Version 2:

Reviewer comments:

Reviewer #1

(Remarks to the Author)

The authors have provided additional data/analysis which can justify their conclusion. I have no more comment and am satisfied about this revision version.

Rebuttal letter “N-acetyl-glucosamine primes *Pseudomonas aeruginosa* for virulence through a type IV pili/cAMP-mediated morphology transition”
Nature Communications NCOMMS-24-49186

We thank the reviewers for their insightful comments that have helped us improve our initial manuscript substantially. We have tried to address their suggestions experimentally whenever technically possible. A detailed reply to each point is to be found below, with our replies written in dark blue.

REVIEWER COMMENTS

Reviewer #1 (Remarks to the Author):

In this manuscript, the authors proposed a new concept that “virulence is acquired through a priming process.” using a *Pseudomonas aeruginosa* infection model. They claimed that priming can be initiated by exposure to extracellular N-acetyl-glucosamine (NAG), which is sensed through the type IV pili (TFP) chemosensor PilJ that activates cAMP signaling. While this story sounds interesting, the conclusion is not very solid based their current experiment settings. I have the following major points:

I) The authors claimed that priming first allows the bacteria to proliferate actively and second allows them to withstand the action of the IMD pathway, likely by resisting the combined antimicrobial action of the two/three AMP families. HigB toxin makes the bacteria vulnerable to a combination of AMPs belonging to two/three families. It is well known that *anr* and *pmr* operons are induced by AMP treatment in *P. aeruginosa*, do these operons get induced *in vivo*? If so, when these operons get induced after infection?

We have measured the expression of these operons by RTqPCR and did not detect any induction of these operons *in vivo* prior to 40 hours (please, see figure below), whereas the bacteria become resistant by 24h as revealed by the transplantation experiments (Fig. 2F) as we did not include these data in our manuscript for the sake of clarity.

Figure 1. Transcription level readout of *anr* operon and *pmr* operons. (A) Transcription level of *anrA* and *PA3559* in PAO1 *in vitro* and in *wt* and *key* flies. (B) Transcription level of *pmrB* and *PA4778* *in vitro* and in *wt* and *key* flies. *wt* flies indicates *P. aeruginosa* infection would trigger antimicrobial peptides production. No AMPs are produced in *key* flies. Primers used for detection are listed below. *anrA*-qPCR-F1: CGACGACCCACGGGAAAACA; *anrA*-qPCR-R1: GCGGATACGCTCCAGCCACA; *PA3559*-qPCR-F: CCCGAAGCCATGCAGGAAAC; *PA3559*-qPCR-R: CCTTGAGCAGCTCGAAATCC; *pmrB*-qPCR-F: CTGTGGCTGAAGCGATGGT; *pmrB*-qPCR-R: TGGCGGGGCTGCGGTAGAAG; *PA4778*-qPCR-F: GTGAAGCGGCGAAGAAAAGC; *PA4778*-qPCR-R: TCGAGGGAGAAAACCGAGGTC

II) The pre-injection of a chitinase into flies minutes prior to a challenge with PAO1 similarly led to enhanced virulence of the injected bacteria. The authors concluded that NAG likely originating from the host promotes the transition to a bacillus morphology, which correlates with virulence. *P. aeruginosa* quorum sensing is known to regulate chitinase production (PMID:11717261). According to this reference “after overnight growth, the majority of the ChiC produced was found intracellularly, whereas only small amounts were detected in the culture medium. However, after several days, the cellular pool of ChiC was largely depleted, and the protein was found in the culture medium.” Thus it is likely a chitinase-like protease is released from *P. aeruginosa* slowly and which is able to generate NAG from host. Do QS get induced during this infection model?

We have addressed this question experimentally and the data are to be found in a novel figure: Fig. S6. We have generated a *chiC* mutant: it presents the same virulence as wild-type *P. aeruginosa* (Fig. S6A). Furthermore, the expression of the *chiC* gene at 40h in wild-type or *kenny* (*key*) flies is actually lower than *in vitro* grown bacteria and the expression of the gene was not detectable at earlier time points (12, 24, and 36 hours) (Fig. S6C).

As regards QS, we did not expect it to be involved in the early infection when priming takes place since we typically inject some 100 bacteria, which do not start proliferating until primed. We have also tested a *lasR* mutant that did not display any altered virulence as displayed below. As regards *mvfR* mutants, they behaved also like wild-type as illustrated below.

Figure 2. Survival curves of flies challenged by $\Delta lasR$ (A) and $\Delta mvfR$ (B).

III) To better understand the role of the *higBA* operon, the authors assessed the quantitative expression of PAO1 proteins in $\Delta higA$, $\Delta higB$, and $\Delta higBA$ bacteria relative to those found in wild-type bacteria. However, the authors only did *in vitro* proteomics analysis as documented in their method part “Wild type PAO1, $\Delta higA$, $\Delta higB$, $\Delta higBA$ mutants (each in biological triplicates) cultured in BHI were harvested by centrifugation at exponential phase and put in dry ice for delivery to the Applied Protein Technology company (Shanghai, China).” Since the authors are able to isolate primed bacteria *in vivo* after infection, why not comparing the proteome of *in vivo* isolated primed bacteria with naïve bacteria of the above strains. We all know *in vitro* and *in vivo* *P. aeruginosa* have distinct transcriptome and proteome (e.g. PMID: 35913171).

We agree with the reviewer that the *in vitro* and *in vivo* transcriptomes and proteomes may differ significantly, a statement that also applies to bacterial physiology as illustrated in this study. However, we faced a technological barrier that results from the limited number of bacteria we can retrieve from infected flies during the early phase of the infection. Even upon death of infected flies, that is when they succumb to bacteremia, we could only collect

10⁷cfu/dying whole fly (and 10⁵-10⁶ from hemolymph), however, we need at least 10¹⁰cfu for each sample. It means that we should collect the hemolymph from 10,000 to 100,000 flies: this would be when the bacteria are already proliferating intensely, that is way after the switch to virulence has taken place. Furthermore, we also tried to send samples of flies infected with PAO1 injection, however, most of the data correspond to host protein and only very few *P. aeruginosa* proteins. We also note that single bacterium transcriptomics as recently reported still needs a substantial number of bacteria, also about 10¹⁰cfus (Blattman *et al.*, Nature, 2024). Indeed, our own attempts to perform *in vivo* bacterial transcriptomics analysis in collaboration with the laboratory of Prof. Marvin Whiteley were a failure.

Nevertheless, the findings derived from *in vitro* proteomics that pinpoint an important role for the T4P downstream of the HigB toxin have been extensively validated *in vivo* through our genetic analysis with rescue of the impaired virulence of *ΔhigA* mutants by *pilJ*, *FimV*, or *cyoB* overexpression, as well as the experiments described in the second half of the manuscript.

VI) The authors suggested that PilJ may be sensing NAG but not NAM. This hypothesis is very weak. Pilus is important for biofilm formation and colonization *in vivo* of *P. aeruginosa*. Type VI pili is also required for full level of QS in biofilm mode of growth. A in-depth biochemical characterization is required to test whether PilJ is able to interact with NAG such as ITC experiment.

This issue has also been raised by Reviewer 2 and we agree that is a highly relevant issue. We have therefore performed an extended set of additional experiments to specifically address this issue experimentally rather than speculating about potential direct sensing of NAG by PilJ as was the case in our initial manuscript (Fig. 4 K-M; Fig. S8 G-H). Briefly, we now additionally show that the shape of *ΔpilJ* mutants remains unchanged in the presence of NAG, in keeping with our survival data that documented that NAG was no longer promoting the virulence of bacteria in *pilJ* mutants (Fig. 4I). We have also tested a direct interaction using the technique of Martin-Mora *et al.* (mBio, 2019) in which these authors had documented a direct interaction between nitrate and the PilJ domain of the McpN chemosensor. Their data was followed by ITC analysis to determine the binding constants of the interaction. As we failed to detect a thermal shift (Fig. 4L) upon the addition of NAG to a recombinant protein containing the two ligand binding domains (LBD) of PilJ, we did not perform ITC analysis. We however performed further genetic experiments of complementation using constructs that lacked the PilJ LBDs and found that these domains are actually dispensable for virulence. We now conclude that the interaction is likely indirect and this situation is reminiscent of the interaction between phosphatidyl ethanolamine and PilJ for directed twitching motility (Kearns, J. Bact, 2001).

V) The authors next examined many genes in the cAMP signaling pathway and the divisome and elongation. This part is not closed linked to the *higBA* operon since their proteomics study does not show significant expression changes of these genes. In fact, the authors written in the introduction “HigA in addition is also able to bind to DNA and functions as a repressor, not only of *higBA* itself but also in the promoters of genes involved in virulence such as *mvfR* or type 3 and 6 secretion systems.” And T3SS and *mvfR* are both published important for infections in different animal models. Thus, again the authors should perform the proteome of *in vivo* isolated primed bacteria with naïve bacteria.

We are not sure that the issues raised here by the reviewer would be solved using proteomics as for the divisome, the relevant information would be the assembly of the complex and its localization within the bacterial cell rather than quantitative changes. As noted above, our attempts at *in vivo* proteomics have been so far unsuccessful as currently mass-spectrometry is not sensitive enough to function on a limited number of bacteria. As also noted in the Reply to

II) above, MvfR and LasR are not involved in virulence in the fly. We had also previously reported that the T3SS system is not required for virulence in *Drosophila* (Limmer, *et al.*, PNAS, 2011). In addition, the proteomics analysis (Fig. 4A) revealed that the T3SS components are more strongly expressed in Δ *higB* mutants whereas they are apparently repressed in Δ *higA* mutants. That T3SS protein level were not changed in the Δ *higB*- Δ *higA* double mutants that are as virulent as Δ *higB* mutants further rules out the T3SS as being a candidate for virulence as described in the Results section on proteomic analysis. Finally, we describe in detail the role of FimV, one of the two most repressed genes found in the proteomics analysis thus establishing a direct link between the *higBA* operon and a protein identified in our proteomic analysis, which plays a capital role in the establishment of the shape of the bacteria as well as in the extent of the elongation of the lateral cell wall.

Reviewer #2 (Remarks to the Author):

I think there are some interesting results in this study, but more experiments and analyses are needed to characterize the morphological change that occurs during infection. There are also several misconceptions about virulence and biofilm formation in *Pseudomonas aeruginosa* that are stated several times throughout the manuscript. Many citations are missing that could better frame results and improve the design of several experiments. Much of the observed data could likely be described via surface-dependent cAMP activation and cellular memory.

We thank the reviewer for providing us with this alternative perspective on our data. We would like to emphasize that we have no evidence for adhesion to the tissues in the systemic infection model, albeit we certainly have an association in our recently described latent infection model (Chen *et al.* PLoS Pathogens, 2024): the association with tissues is however not dependent on PilJ (unpublished data). We have also no evidence for the formation of biofilms (in more than 15 years of working on *P. aeruginosa* we just saw once a fly that seemed to have a biofilm forming in the hemocoel of the fly). Thus, biofilm does not appear to be a relevant parameter in the current model of infections in *Drosophila*.

As much of the pathology in the systemic infection model seems to be linked to bacteremia in the hemolymph rather than direct damages of tissues, we had not considered the possibility of a transient interaction with host surfaces. We acknowledge that we cannot formally exclude this hypothesis. While we had read a number of articles on surface sensing by bacteria, the Lee *et al.* (PNAS, 2018) article on “cellular memory” had escaped our attention and indeed this hypothesis is worth considering. We therefore addressed this issue experimentally in two ways. First, we asked whether having bacteria associate with an agarose surface *in vitro* would prime the bacteria for virulence. As shown in Fig. S8D-E, even though we did get some transcriptional activation (actually increase of steady-state mRNA levels) of *xphA*, we did not observe any priming. Second, we have generated mutants that would affect T4P formation and not the chemosensation apparatus, namely *pilA* and *pilC*. While these mutants are expected to abrogate surface sensing, they nevertheless displayed only a slightly impaired virulence that is clearly not of the range that we observed in *pilJ* mutants (Fig. S8F). Finally, in their pioneering analysis of twitching motility mutants, D’Argenio *et al.* (J. Bact, 2001) had already made the point that twitching motility and virulence were separable and had described multiple mutants, including *pilY1* and *pilT*, for which twitching motility was affected without influencing virulence. Taken together, these data demonstrate that the priming phenomenon we report in our manuscript is independent from surface sensing and have been included in the manuscript.

We have now referenced in the text more articles dealing with surface sensing of bacteria.

-lines 49-53: Both acute virulence and biofilm formation require surface contact and surface sensing via Type IV pili in *Pseudomonas aeruginosa*. Surface sensing triggers cAMP activation via type IV pili interaction with the surface leading to activation of the global virulence regulator, Vfr, which is critical for expression of T2SS and T3SS PMID: 26041805, 37382531, 20494996, and many others.

Virulence is primarily defined by the impact that bacterial infection has on the survival of the host in a given infection model. For instance, in the *Drosophila* model, we did not find any role for the T3SS in virulence (Limmer *et al.*, PNAS, 2011). In our initial manuscript, we found that *vfR* overexpression did not rescue the *pilJ* impaired virulence phenotype. Upon thinking about this reviewer's comments, we agree that the overexpression of Vfr might not be sufficient to initiate its activation, for instance if cAMP levels remain too low to activate the overproduced protein, as suggested by the reviewer (see further below). We therefore decided to generate a ΔvfR mutant and analyze its virulence. As now shown in Fig. 4E-F, we found that the ΔvfR mutant displays a level of impaired virulence that is similar to that of *ApilJ*. Importantly, we also found that these bacteria remained ellipsoid *in vivo* (Fig. 4G-G'). We conclude that contrarily to our superficial initial conclusions, Vfr does indeed play an important role for the virulence of PAO1 in our systemic infection model and gratefully acknowledge the suggestion of the reviewer that clarified the issue.

Type IV pili interaction with the surface also triggers c-di-GMP production leading to biofilm formation PMID: 25626906, 34168081. Furthermore, type IV pili are critical virulence factors and allow for motility within epithelial cells likely triggering cAMP production PMID: 31431558. cAMP activation due to type iv pili surface sensing has also been shown to persist for multiple generations within *Pseudomonas aeruginosa* and serve as a molecular memory of surface contact which explains the continued virulence of cells after being recovered from the first infection PMID: 29559526. Also, the extent of cAMP signaling relates to the stiffness of the material to which type IV pili are attached to PMID: 35561220, which is a maximum when binding to surfaces with a stiffness similar to that of human tissues.

We have already discussed above many of the points raised by the reviewer in this paragraph. As regards c-di-GMP production, it depends on PilY1 (Luo *et al.*, mBio, 2015). As noted above, D'Argenio *et al.* (J. Bact, 2001) have already tested this mutant and found it to display a normal virulence.

-For all figures reporting CFU data, please change the y-axis to log₁₀ CFU/ml.

Done

-Fig1. Instead of adding 1 to your Cfu data to avoid you can state "below limit of detection" or "0 CFU".

-lines 106-107: the important control here is that *higA* and the *higB* OE strain grow at the same rate as WT. Growth curves are typically done via 24-hour growth curves with many time points characterizing each phase of the growth curve, not 5 time points over five days. There is no description of the *in vitro* growth conditions in the methods.

We have now described our *in vitro* growth conditions in the Methods section. We have included in the manuscript this long-time course as it reflects the actual conditions *in vivo* in terms of temperature and length of the experiment. We include below the data for 14-hour growth curve at 37°C that did not reveal any major difference in the growth curves, except for

the $\Delta higB$ mutant that however displays an enhanced virulence *in vivo*, like the $\Delta higB-\Delta higA$ mutant that grows like wild-type PAO1.

Figure 3. Growth curves for the set of *higBA* mutants compared with wild type PAO1 at 37°C. The density of the bacterial solution was measured by spectrophotometer at OD₆₀₀.

-lines 136-138 reference "a lack of bacterial growth until at least 18h (Fig. S3E)," but fig S3E is a survival curve of flies and does not show an increase in bacterial CFU over time which would be indicative of growth.

We thank the reviewer for pointing out this sentence that was not perfectly clear. The text now reads: "The finding that $\Delta higA$ recovers its virulence in flies missing either IMD pathway receptors PGRP-LC and PGRP-LE, which senses bacterial growth (Fig. S3E), further supports a lack of bacterial growth until at least 18h (Fig. 1B), although we cannot formally exclude an active inhibition of the IMD pathway by injected PAO1¹⁰" (the IMD pathway is triggered upon sensing polymeric PGN or short PGN degradation fragment such as tracheal cytotoxin. Because polymeric PGN in INTACT bacteria is buried under the outer membrane, it is not accessible to the PGRP-LC transmembrane sensor. The cell wall remodeling that accompanies cell division leads to the release of short PGN fragments that are sensed both by PGRP-LC specific isoforms and PGRP-LE intracellularly. Hence, the IMD pathway gets activated only upon bacterial cell division).

-lines 159-166: PMID: 29559526

We understand what the reviewer had in mind, that is, that the multigenerational memory of "priming" to surfaces might account for the priming effect we describe in this experiment. As, discussed above, we have now rigorously tested the hypothesis of surface sensing in two independent manners and have excluded this possibility.

-line 167: please explain how "priming" is different or distinct from any other typical lag phases in a new growth medium.

Priming is distinct from a typical lag phase in that it is required only in wild-type but not immunodeficient *kenny* flies. Also, we are not aware that there are morphologic changes of the bacteria when transferred to a novel growth medium. Finally, some pathogens such as *Serratia marcescens* kill flies in less than a day upon the injection of only five bacteria (Nehme *et al.*, *PLoS Pathogens*, 2007).

-line 175-180/Fig3A-C: There needs to be some sort of quantification with some sort of image analysis software to conclude that there is any significant difference in cell shape/morphology. Without any sort of quantification and statistical analysis I am not comfortable making any sort of conclusion. Reference 9 also lacks any sort of statistical image analysis of the morphological changes of bacteria. Furthermore, if the shape of bacterial cells are to be evaluated, they should all be in the same focal plane.

We have now added EM data and associated quantification of length and length/width ration (Fig. 3A-A''). As regards light microscopy, we have focused on measuring the length of the bacteria for several key experiments (Fig. 3B', 3D'; 4G', 4K', 5B', 5C', 5F', and 5J').

We are measuring the lengths of bacteria as observed through the cuticle of the fly and few bacteria per fly are "measurable", hence making quantification challenging. Please, note that the figures displaying bacterial shapes are "collages" in which we have pieced together in one panel pictures taken from different fields that do not come necessarily from the same fly, as noted in the figure legends. For each bacterium, we took the pictures at the optimal focal plane as much as possible.

-all overexpression experiments should include westerns to show stable overexpression of the protein.

We agree with the reviewer that it ideally should be done. However, we certainly do not have access to antibodies for any of the constructs we have made and therefore we cannot use Western blot analysis (it is administratively very difficult to import biological reagents into China). Yet, in a number of cases, we do observe a biological effect that indicates that the overexpression construct is functional (*fimV*, *pilJ*, a number of cell division genes such as *ftsK*, *ftsQ*, *ftsW*...). In addition, we note that we can rescue the *pilJ* mutant using the *pilJ* overexpression construct (Fig. 4M). We have also made GFP fusions for FimV, which we can directly observe *in vivo* and its deletion constructs as it is a key gene in our analysis (Fig. 5). Thus, we are confident that our main conclusions are correct. We have found a number of cell division genes that do not produce a phenotype when overexpressed (*ftsL*, *ftsI*, *ftsX*, and *ftsE*). As these genes are not critical for our analysis, we have simply monitored that they were indeed overexpressed using RTqPCR (Fig. S10G). We acknowledge in the Supplementary Figure Legend that the overexpressed proteins might be unstable to underscore the potential limitation of this experiment that yielded negative results.

-Figure S5E: there seems to be a significant difference between Nam_higA strain w/ 0mg/ml and 10mg/ml but there are not statistical tests to tell if anything is different.

There was indeed a problem with the 0 mg/mL *ΔhigA* data points, which should have given similar values for NAM and NAG (no addition). The "no NAM" sample is abnormally low due to water evaporation in the three wells of the 96-well plate. However, since no major differences have been observed for the eleven NAM/NAG concentrations tested, we do not think this represents a significant issue. Of course, should the reviewer imperatively ask for it, we will repeat the experiment.

-Fig 4C: please confirm the overexpression of PilJ with a western.

While we do not have an antibody against PilJ, we have performed genetic rescue experiments with the full or domain-deleted genes and have found that the full gene (and the LBD deletions) are able to rescue the *ΔpilJ* mutant phenotype (Fig.4M).

-Line 271: One of the few ligands that actually bind to the ligand binding domain of PilJ are phenol soluble modulins produced by Staph aureus. To properly test if PilJ is sensing NAG, you should use PilJ with the ligand binding domain removed which has been used in several studies so far: PMID 27190147, 38349934. Without this control you cannot conclude that PilJ is sensing NAG and not just transducing a surface signal from type IV pili.

We thank the reviewer for the suggestion and have performed the suggested experiments and additional ones. Namely, we have addressed the issue of direct binding of NAG to the PilJ RBD domains both biochemically and genetically (see also reply to Reviewer 1 Point IV). The new data are reported in Fig. 4 K-M and Fig. S8G-H. We have not altered our conclusion: the data show that the RBD domain is dispensable for the action of NAG, which fails to bind to the RBD domains *in vitro*. We also compare our results with those of Kearns *et al.* (J. Bact, 2001) as regards directed twitching motility towards phosphatidylethanolamine, which also does not involve the PilJ RBD domains.

-line 277: the citations listed do not connect cAMP levels with cell wall remodeling

We have modified our wording of this sentence to : “cAMP levels control bacterial cell morphology and also activate the Vfr transcription factor, which regulates the expression of several virulence genes^{4,30,32-35}.”

-lines 273-287: Even though overexpression of vfr does not compensate virulence it does not mean Vfr does not have a downstream role in virulence given its function. Vfr has a cAMP dependent and cAMP independent regulon PMID: 20494996. Overexpression of Vfr without an equal activation of cAMP means it is only activating the non-cAMP regulon which are not virulence genes. Furthermore, Vfr is the only known cAMP binding protein in Pseudomonas with any physiological effect. PMID 38832786. The other mutants indicate Vfr likely plays a role in a cAMP-dependent manner.

We agree with the reviewer, who is right, and have addressed experimentally this concern (please, see reply to lines 51-53 above).

-Fig 5E: There should be quantification of polar localization like using microbeJ or Fiji. FimV appears to be cytoplasmic when OE in liquid conditions. Surface sensing leads to an increase in the number of type iv pili as well as localization of pili components.

We do not feel that image analysis is required to determine the polar localization of FimV-GFP as the human eye is certainly faster. We do however agree that quantification is desirable and have performed it. We have amended our text to mention the quantitative aspect of FimV-GFP localization *in vitro* and *in vivo*: “Strikingly, whereas tagged FimV was found distributed rather evenly throughout the cell internal surface in 60% of cells grown in BHB, it was found to be enriched at the poles of all bacteria injected into wild-type flies 24h after challenge (Fig. 5C-C’), which suggests that bacterial pole organization changes as a result of priming.”

-line 313-316: The data do not support the claim made. These mutations are supposed to make the cell morphology more rod like and therefore virulent but in line 308-312 you state that overexpression leads to decreased virulence.

We think there is a misunderstanding here. The statements mentioned by the reviewer actually apply to *ftsZ* overexpression **in a *higA* mutant and not in a wild-type background**. The bacteria are rounded as shown in what is now FigS10A and reported in Table S2. The

overexpression of *ftsZ* in wild-type PAO1 leads to an elongated shape *in vitro* as reported in the literature but to an ellipsoid to rod shape *in vivo*, as documented also in Fig. S10 A and Table S2. In this case, the virulence of PAO1 is only mildly altered (Fig. S10D).

-Type IV pili interact with surfaces and are able to bind surfaces better with EPS. Perhaps NAG can bind type iv pili and mimic a surface that would trigger Pil-Chp, leading to cAMP production, Vfr-cAMP dependent regulon, and increased virulence. This positive feedback loop maintains high levels of cAMP over multiple generations and is a hypothesis worth testing here.

While we understand to some degree the line of reasoning of the reviewer (we have difficulty though visualizing how NAG, which are not polymers, could form a surface), our experiments that directly tested the hypothesis of surface sensing do not support this possibility, especially the finding that *pilA* mutants have a normal virulence. However, the reviewer is right in the sense that NAG is not sensed directly by the PilJ tandem ligand-binding domains and do not currently understand exactly how it is perceived by bacteria through PilJ.

-line 352: mechanosensing is not tested in this manuscript so this claim cannot be made.

We now have tested in two independent manner this hypothesis and believe it is now disproved.

-line 355-357: many papers have already shown this to not be true.

We have now clarified the statement that applied to the study of host-pathogen interactions in the *Drosophila* field: “In contrast to the prevailing view of immediate virulence of pathogenic bacteria in *Drosophila* systemic infection models, *P. aeruginosa* virulence in the host requires an adaptation period”.

-line 383-387: The claim about bacterial shape and resistance to the IMD pathway is not supported by the last set of experiments in which mutants exhibiting more rod-like morphology are still cleared by IMD.

We are not sure we understand the reviewer’s point here that deals with the Discussion. Is it a reference to the comment above on lines 313-316 that we have addressed? Or is the reviewer referring to: “Strikingly, bacteria overexpressing *zipA* or *ftsW* formed filaments *in vitro*, yet displayed a rather rounded to ellipsoid morphology when injected into wild-type flies, yet was rod-shaped when injected into *key* flies (Fig. 5J-J’).” As shown in Fig. 5J, even though the *ftsW* or *zipA* overexpressing bacteria filament *in vitro*, they do not *in vivo*. They are rounded in wild-type flies but present a mixture between ellipsoid to rod morphology in *key* mutants. While addressing this comment, we have noted a mistake in Table S2 that we have now corrected as it was incorrectly referring to a rounded to ellipsoid morphology of these bacteria in *key* mutants when they are rod to ellipsoid morphologies. We do observe a rod morphology upon *fimV* overexpression in wild-type bacteria, which corresponds to a slightly impaired virulence in wild-type flies, likely mirroring a degree of toxicity of the construct since the virulence was also mildly impaired in *key* mutants (Table S2).

-FimV is necessary for cAMP activation via type iv pili and flagellar mediated surface sensing so it makes sense increased FimV increases virulence phenotypes. PMID: 31682637

We agree with the reviewer. Our point with respect to cAMP signaling however is that FimV overexpression has much stronger phenotype than *pilJ* overexpression, implying that FimV has other function independent of cAMP. We now propose a model in which FimV is important in the determination of cell shape and virulence by providing a polarity information to the bacterium that favors lateral cell wall extension and that cAMP signaling is equally important in conveying a signal that enhances lateral cell wall synthesis, as now added in the Discussion and our dissection of the FimV domains in relation to cell shape and virulence.

Reviewer #3 (Remarks to the Author):

The present work describes the process by which the bacterial pathogen *Pseudomonas aeruginosa* adapts to the host environment in the initial stage of the infection. Using *Drosophila melanogaster* as their acute infection model, this transition period allows the bacteria to not proliferate, hiding from the host's immune system, and switching from an ellipsoid shape to a rod morphology. The work clearly shows a correlation between the bacillus morphology and virulence. Furthermore, the authors also elucidate the probable mechanism, from the extracellular signal required to initiate the transition to the genes related to bacteria wall remodeling. Overall, the data is solid with some minor concerns over significance in terms of extrapolating to mammalian infection models. The authors should address this in more detail in the discussion especially in light of the use of antimicrobial peptides.

We have followed the reviewer's suggestion and expended the last paragraph of the Discussion. It is clear that the easiest model to address in mammals the link between bacterial shape and virulence as described in our work would be to use a wounding model, likely the burn wound model. While there are over 500 publications dealing with this model retrievable in PubMed, we have not read most of them, looking at the Abstracts of several studies by adding key words related to bacterial shape or morphology or T4P did not retrieve relevant references and the impression is that investigators look at time points that are too late, that is 24 to 48 hours of inoculation. Clearly, a study of the type we have pursued in *Drosophila* is warranted in a mammalian burn wound model (mouse, rat, pig? Mice would likely be the cheapest way to go). We have cited the original study as well as another one that underscores the importance of wounding the fascia: would this lead to the release of NAG?

Despite this the work clearly helps expand the understanding about the molecular mechanisms involved in pathogen adaptation to the host during different stages of infection, here specifically during the acute stage. This knowledge is crucial to understand the process of colonization that results from bacterial infections and for the development of new therapies and treatments targeting bacterial pathogens characterized as multi-drug resistant, such as *Pseudomonas aeruginosa*.

The design and analysis of a wide variety of mutants in *P. aeruginosa* and on the side of the host (flies) is an innovative allowing a mechanistic teasing out of the role of the NAG in remodelling cell morphology and priming of virulence. They clearly show the sensing of NAG is a common signal that promotes virulence through morphology changes. However, the authors might consider higher level imaging such as electron microscopy to confirm the morphology experiments as it is difficult to really judge the morphologies.

As described above for Reviewer 2, we have now added additional EM data in Fig. 3 and have quantified the length of bacteria in the key experiments of the article. Importantly, besides the

rod shape, it seems that the absolute length of the bacteria is also an important parameter in for the virulence of bacteria through their ability to withstand the action of three AMP families regulated by the IMD pathway.

Also, in Fig 3A-C more detail of the conditions and time frame for the images would be helpful to the reader.

This remark has been addressed.

The priming phase by NAG is mediated by the T4P chemosensing system by activating downstream signaling via cAMP mediated cell wall remodeling. The authors make the case that morphology drives virulence. Overall the technical quality of the manuscript is extremely high and the results consistent with the conclusions.

Although the experiments as performed are of a high quality the text needs to be improved in consistency and clarity. Specifically, I encourage the authors to review the consistency between the names listed in the tables and the text and figures, as some of them do not match or are not shown in the tables.

We hope this concern has been addressed.

Then, I would recommend including a complete list of the plasmids designed and/or assayed as well as for the bacteria or fly mutants. In addition to just the reference, including a brief description of the constructs, such as the plasmid backbone, inducible or constitutive promoter, gene/pathway affected, etc, would be recommended and would improve the understanding of the text.

Done, with new supplementary Tables.

Overall, this is a rigorous and interesting report on *P. aeruginosa* priming and adaptation in the *D. melanogaster* model system. The results are significant in providing a foundation for the role of priming in an acute infection and open further avenues of research in other model systems.

Reviewer #4 (Remarks to the Author):

We do hope that the process was instructive.

Rebuttal letter (second round of revisions)

We thank the reviewers for their generally positive comments and have addressed experimentally some of the points raised by Reviewer 1.

Reviewer #1 (Remarks to the Author):

The study investigates the role of N-acetyl-glucosamine (NAG) in priming *Pseudomonas aeruginosa* for virulence via a T4P/cAMP-mediated morphological transition. While the revised manuscript addresses some concerns and presents novel insights into host-pathogen interactions, several critical issues remain unresolved. Below are detailed critiques to strengthen the scientific rigor and clarity of the work:

Major Concerns

1. Image Quality and Presentation

- Several fluorescence microscopy images lack clarity (e.g., Fig. 3C, 3D, 4G, 5B–C). The use of composite panels from disparate fields creates confusion, and scale bars are inconsistently placed or absent. The authors should provide single-field, high-resolution images with uniform scale bars (e.g., 2 μm) clearly labeled in all panels. Also, please include detailed figure legends explaining image stitching (if necessary) and statistical significance (e.g., n-values, replicates). Standardize annotations are also required (e.g., replace “wt” with “WT,” remove underscores in labels like “WT_?”).

We fear that this reviewer is not familiar with working on imaging bacteria in living tissues. It is totally different from imaging *in vitro*. One important limitation is that there are few bacteria that we can image in tissues given their low numbers, especially at early phases of the infection. When several bacteria are present in the same field, they are not on the same focal plan when working at high magnification. **THUS, IT IS NOT POSSIBLE TO AVOID STITCHING IF ONE WANTS TO AVOID SHOWING JUST A SINGLE BACTERIUM PER PANEL.** The senior author of this study has been a *Drosophila* developmental biologist last millennium and would never have thought that it would be possible to get such high-quality images in *Drosophila* tissues (we usually use a 63x immersion objective for imaging *Drosophila* embryos and working with a 100x objective is especially difficult). We have often used the staining of LPS-O-antigen to visualize bacterial shape, which is by essence somewhat fuzzy.

We have nevertheless attempted to improve the quality of images. One technique was to image bacteria retrieved from hemolymph as it makes it a bit easier to find the bacteria and to quantify them using Microbe J. Of note, as noted in the Methods section, we have checked that bacteria retrieved from the hemolymph display shape properties similar to those observed in tissues as we might have introduced a bias by using such a technique, for instance by selecting bacteria not associated with tissues. We have also increased the number of bacteria analyzed by Microbe J, which required an important work load since bacteria have to be imaged one by one.

Figure legends have been improved as well as scale bars.

2. Mechanistic Evidence for NAG Sensing by PilJ

- The claim that PilJ indirectly senses NAG is weakly supported. While the thermal shift assay (Fig. 4L) shows no direct binding of NAG to the PilJ LBD, no alternative mechanism is proposed. The authors should measure cAMP levels in ΔpilJ mutants exposed to NAG to test whether NAG activates cAMP signaling in a PilJ-dependent manner. They should perform genetic complementation with PilJ truncation mutants (e.g., ΔLBD) to confirm if NAG sensing requires other domains (e.g., transmembrane or HAMP domains).

We have monitored cAMP signaling by measuring the steady-state transcripts of *pilA*. This analysis revealed that cAMP signaling is switched on *in vivo*, independently of the presence of NAG (Fig. 5f, j, k). We had already performed the rescue of ΔpilJ by a *pilJ* construct lacking the LBD domains, it did rescue, and also of the tandem LBDs alone. We have now additionally

shown that co-injecting NAG with the rescued *pilJ* by the construct lacking LBD domains did enhance the virulence of the injected bacteria, further demonstrating that the LBD domains are dispensable for NAG action on virulence *in vivo*. The further delineation of the PilJ domains mediating the effects of NAG is outside of the scope of this study.

3. Lack of Validation in Mammalian Models

The study relies solely on *Drosophila* infection assays. The relevance of the proposed NAG/PilJ/cAMP/FimV axis in mammalian models (e.g., murine acute lung infection) remains unaddressed. They could test virulence of Δ *higA*, Δ *pilJ*, or NAG-treated PAO1 in a mouse model (e.g., intratracheal infection). If animal experiments are infeasible, explicitly discuss the limitations of extrapolating findings from flies to mammals.

We have discussed this issue at the end of the Discussion. A more in-depth discussion would unnecessarily lengthen the manuscript. This issue should be discussed more extensively in a Commentary article. We also note that the number of studies documenting the bacterial shape in animal tissues at our level of resolution is low, if not null. The onus is now on investigators working with mammalian models to validate our findings, for instance in burnt-wound models.

4. Absence of Protein-Level Validation

The key molecular mechanisms (e.g., FimV localization, PilJ expression in truncation mutants) lack validation via Western blotting or immunostaining. The authors should confirm FimV protein expression and polar localization using anti-FimV antibodies (commercially available or custom-designed). They should validate expression levels of truncated PilJ constructs (Fig. 4M) via Western blot to ensure genetic complementation is functional.

This issue had already been raised by two other reviewers and our reply was deemed satisfactory (please, see Reviews to first Round of revisions). The best evidence is that the FimV-GFP fusion has biological activity similar to that reported for unlabeled FimV, that is an elongation of bacteria. We have demonstrated that the overexpression of full *pilJ* strongly rescues the Δ *pilJ* mutant phenotype, which genetically demonstrates that the construct is functional, a degree of evidence that is much stronger than a Western blot: the addition of a tag can kill the biological activity of a protein, yet look very nice on a Western blot. That NAG still promotes the virulence of the rescue by the form lacking the LBD domains demonstrates that this truncated form is also functional, a major result validating this construct.

5. Quantification of Morphological Data

The central claim linking bacillus morphology to virulence relies on qualitative images (e.g., Fig. 3A–C, 4G–K) without robust quantification. The authors could use automated tools (e.g., CellProfiler, MicrobeJ) to quantify bacterial length, width, and aspect ratio across ≥ 100 cells per condition, and include statistical comparisons (e.g., ANOVA with post-hoc tests) between groups (e.g., WT vs. Δ higA in key flies).

Please, see Reply to Point 1. Of note, we have increased the number of quantified bacteria as much as possible. Since the differences are statistically significant, it makes no scientific sense to quantify more bacteria to reach this artificial threshold of 100, which may be easy to reach when working *in vitro*... Of note, when we perform survival experiments, we use usually between 120 to 180 flies per condition. Yet, when reviewing articles, I do not request that investigators working on mice use a similar number of animals...

We have changed the pictures of Fig. 3b-c, Fig. 4g, Fig. 5b, Fig. 6b, c, f, j, Fig. S9d.

We have increased the number of bacteria quantified: Fig. 3b', d', Fig. 4g', Fig. 5b', Fig. 6b', f', j', Fig. S5a (quantification added), Fig. S9d'-d'''' (quantification added, including width). With respect to new data, bacteria retrieved 10h after co-injection with NAM or NAG, we have also monitored width to document that NAM-treated bacteria are more rounded *in vivo*, in keeping with a loss of polarity.

6. Fitness of Overexpression Strains

The partial rescue of Δ higA virulence by pilJ, cyaB, or fimV overexpression (Figs. 4C–D, 5A) may reflect fitness trade-offs rather than mechanistic effects. The authors should perform growth curves or competitive index assays *in vitro* and *in vivo* to rule out fitness costs.

This analysis has actually been performed systematically from the very first manuscript by monitoring the virulence of bacteria in *key* immune-deficient flies in which there is no AMP expression that acts on injected bacteria. We have always pointed out when the *in vivo* fitness of injected bacteria appeared to be somewhat lessened: Fig. 1e, g, Fig. 2f, Fig. 3b, d-h, Fig. 4b-d, g Fig. 5a, b, d, e, g, Fig. S8d (for Fig. 5j), Fig. 6a, b, e, h, i, j, Fig. S2f, Fig. S5b, c, e, f, h, i, Fig. S6b, Fig. S6g, Fig. S7e, f, Fig. S9a-b, e, Fig. S10b-f.

7. Unresolved Mechanism of NAM Toxicity

The observation that NAM abolishes virulence (Fig. 3I) is intriguing but unexplained.

We have now made a major step toward identifying the mechanism of NAM toxicity. It prevents the association of FimV to the poles, likely by preventing the binding of FimV to the septal peptidoglycan, as now mentioned in the revised Discussion. What is fascinating is that NAG counteracts the effect of NAM to some extent, as inferred from survival analysis. Strikingly, NAG does promote a faster localization of FimV-GFP to the poles (Fig. S9h-h'). Further structural elucidation of a molecular mechanism by competition between NAM, but not NAG, and septal peptidoglycan for a peptidoglycan binding site in FimV lies outside of the scope of this study.

Of note, the rescue of the Δ higA by the FimV truncated construct lacking the LysM domain (a peptidoglycan-binding domain) takes place in a context in which there is still some endogenous full-length FimV expression that is likely enough to nucleate low quantities of endogenous FimV to the poles, which may allow the subsequent initiation of a weak bacterial polarity through protein-protein interactions able to compensate the absence of the LysM domain in overexpressed FimV. This may explain why the LysM truncated construct is still able to rescue rather well the Δ higA mutant.

Minor Concerns

1. Figure Standardization

-Color schemes: Adopt consistent color coding across all figures (e.g., use Fig. 1B's orange/green/blue/red scheme for WT/mutant/rescue groups).

- Axis labels:

- Remove redundant labels (e.g., "10" in Fig. 1A x-axis).

- Use uniform terminology (e.g., "Bacterial load (CFU/fly)" in Fig. 1B, 2C).

- Fonts and symbols:

- Standardize font sizes and styles (e.g., Fig. 3B' uses "PAO1" in a different font).

- Replace "ns" with "n.s." (not significant) and ensure asterisks align with statistical thresholds.

2. Scheme Clarity

- Simplify and standardize schematics (e.g., Fig. 3J, 4H, 5G). Move overly simplistic diagrams to supplementary materials.

3. Technical Details

- Clarify ambiguous labels (e.g., "rpl49" in Fig. 2A should be defined as a normalization control).

- Correct distorted fonts (e.g., Fig. 4L') and align panel labels (e.g., "WT" vs. "key" in Fig. 3B').

Done.

Reviewer #2 (Remarks to the Author):

The authors have performed a substantial amount work and quantification to improve the manuscript and have adequately addressed each of my concerns. Below are minor comments:

-Recent studies have shown a significant difference in surface sensing, colonization, and biofilm formation between PAO1 and PA14, specifically with Wsp/c-di-GMP being much more important for PAO1 and Pil-Chp/cAMP being more important for PA14 PMID: 34516283, 32098815. This is one of the first studies to highlight Pil-Chp in PAO1, and I wonder how these experiments would differ if PA14 were used, although this is out of the scope of this study.

We have actually reproduced some of these data ($\Delta fimV$ and Δvfr) using the PA14 strain. We have not included these data in the present study.

-line 321-324 change PilJ gene to lowercase

Done.

Reviewer #3 (Remarks to the Author):

The authors have performed a significant number of extra experiments to address the major concerns including extra qPCR experiments to ensure and and per are not induced on AMP treatment, generation of a chic mutant that has a similar phenotype to WT indicating the NAG is not generated by the host proteases. Moreover, they tested the cAMP activation and memory with additional experiments to show priming is affected independent of surface sensing. Overall the manuscript is vastly improved and contributes to our understanding of virulence priming by NAG.

Reviewer #4 (Remarks to the Author):

We thank all reviewers for their inputs and suggestions to improve the quality of the manuscript.